# Groundwater data improve modelling of headwater stream $CO_2$ outgassing with a stable DIC isotope approach

Anne Marx [1], Marcus Conrad [2], Vadym Aizinger [3,2], Alexander Prechtel [2], Robert van Geldern [1], Johannes A.C. Barth [1]

[1]Department of Geography and Geosciences, GeoZentrum Nordbayern, Friedrich-Alexander-University Erlangen-Nürnberg (FAU), Schlossgarten 5, 91054 Erlangen, Germany

[2]Department of Mathematics, Friedrich-Alexander-University Erlangen-Nürnberg (FAU), Cauerstr. 11, 91058 Erlangen, Germany

[3]Computing Center, Alfred Wegener Institute, Helmholtz Centre for Polar and Marine Research, Am Handelshafen 12, 27570 Bremerhaven, Germany

*Correspondence to*: Anne Marx (anne.marx@fau.de)

**Abstract.** A large portion of terrestrially derived carbon outgasses as carbon dioxide ($CO_2$) from streams and rivers to the atmosphere. Particularly, the amount of $CO_2$ outgassing from small headwater streams was indicated as highly uncertain. Conservative estimates suggest that they contribute 36 % (i.e., 0.93 petagrams C $yr^{-1}$) of total $CO_2$ outgassing from all fluvial ecosystems on the globe. In this study, stream $pCO_2$, dissolved inorganic carbon (DIC) and $\delta^{13}C_{DIC}$ data were used to determine $CO_2$ outgassing from an acidic headwater stream in the Uhlirska catchment (Czech Republic). This stream drains a catchment with silicate bedrock. The applied stable isotope model is based on the principle, that the $^{13}C/^{12}C$ ratio of its sources and the intensity of $CO_2$ outgassing control the isotope ratio of DIC in stream water. It avoids the use of the gas transfer velocity parameter ($k$) that is highly variable and mostly difficult to constrain. Model results indicate that $CO_2$ outgassing contributed more than 80 % to the annual stream inorganic carbon loss in the Uhlirska catchment. This translated to a $CO_2$ outgassing rate from the stream of 34.9 kg C $m^{-2}$ $yr^{-1}$, when normalised to the stream surface area. Large temporal variations with maximum values shortly before spring snowmelt and in summer emphasise the need for investigations at higher temporal resolution. We improved the model uncertainty by incorporating groundwater data to better constrain the isotope compositions of initial DIC. Due to the large global abundance of acidic, humic-rich headwaters, we underline the importance of this integral approach for global applications.

## 1 Introduction

Rivers and streams are the main carbon pathways from the continents to the oceans and thus constitute an important link in the global carbon cycle. In the process of transport, large amounts of carbon – mostly in the form of $CO_2$ – outgas from the water surface to the atmosphere (Cole et al., 2007;Aufdenkampe et al., 2011;Regnier et al., 2013;Wehrli, 2013). Globally, this form of $CO_2$ contributions to the atmosphere was estimated between 0.6 to 2.6 petagrams (Pg) of carbon per year (Raymond et al., 2013;Lauerwald et al., 2015;Sawakuchi et al., 2017;Marx et al., 2017a). The lower value of this range by Lauerwald et al. (2015) excluded streams of Strahler stream numbers below three. This is because of sparse coverage of actual direct measurements of the partial pressure of $CO_2$ ($pCO_2$) in headwater streams. Note that the upper value of the

range still lacks a representative contribution from headwater streams (Marx et al., 2017a).

The contributions of headwater streams are considered as a major unknown factor in these estimates of global carbon budgets for inland waters (Cole et al., 2007;Raymond et al., 2013). The main reasons are uncertainties of groundwater input as well as poorly defined surface areas and gas transfer velocities (Marx et al., 2017a;Schelker et al., 2016). In addition,

$p$CO$_2$ and subsequent CO$_2$ outgassing fluxes typically decline rapidly from stream source areas to river sections further downstream (van Geldern et al., 2015;Stets et al., 2017). Poor definition of these gradients adds another uncertainty to the global carbon budget. The enormous number of small headwater streams making a significant contribution on a basin and thus on the continental scale combined with the scarcity of data led to the term *aqua incognita* (Bishop et al., 2008).

Various direct and indirect approaches to determine CO$_2$ fluxes exist. Most often fluxes are calculated from $p$CO$_2$ and gas

transfer velocities (Teodoru et al., 2009;Raymond et al., 2012;Lauerwald et al., 2015;van Geldern et al., 2015). However, because of large variabilities of gas transfer velocities on different spatial and temporal scales, this type of determination remains controversial, especially for small-scale applications (Marx et al., 2017a;Regnier et al., 2013;Schelker et al., 2016). For applications in small streams and during changing flow conditions also direct methods such as floating chamber approaches exhibit major drawbacks such as altered outgassing behaviour because of artificially created currents inside

anchored chambers (Lorke et al., 2015;Bastviken et al., 2015). In addition, rapid downstream losses of CO$_2$ often imply that CO$_2$-rich groundwater inputs are lost before actual measurements can take place (Reichert et al., 2009).

Recent approaches have used stable carbon isotopes of dissolved inorganic carbon to reliably quantify CO$_2$ outgassing from streams and rivers (Polsenaere and Abril, 2012;Venkiteswaran et al., 2014), in which the model by Venkiteswaran et al. (2014) is a parsimonious, simpler version of the model by Polsenaere and Abril (2012). Both apply inverse modelling to

calculate the amount of CO$_2$ lost upstream of a sampling point within a stream or at a catchment outlet. One clear advantage when compared to conventional methods is, that these stable isotope approaches account for the potentially high CO$_2$ outgassing upstream of any sampling point. Moreover, they incorporate groundwater seeps in first-order headwaters, particularly at low discharge (Polsenaere and Abril, 2012). These factors are typically not covered by conventional methods.

The integrative models exploit the fact, that the stable isotope ratios of dissolved inorganic carbon (expressed as $\delta^{13}$C$_{DIC}$) in

stream water are controlled by $^{13}$C/$^{12}$C ratios of its sources and the intensity of CO$_2$ outgassing. One important input is the stable isotope ratio of soil CO$_2$ that in turn depends on the plants' pathways used for photosynthesis and the organic matter sources fuelling plant and microbial respiration (Mook et al., 1983;Vogel, 1993). In general, this soil-internally produced CO$_2$ has a $\delta^{13}$C$_{CO_2}$ value close to the initial substrate, which has a range from –30 to –24 ‰ for the most commonly occurring C3 plants (Ehleringer and Cerling, 2002). After entering the stream, CO$_2$ outgassing to the atmosphere increases

$^{13}$C/$^{12}$C ratios in the remaining DIC pool because of the well-known equilibrium isotopic fractionation between CO$_2$, HCO$_3^-$, and CO$_3^{2-}$ (Myrttinen et al., 2015, 2012;Mook et al., 1974). This predictable and temperature-related process is calculated by the models of Polsenaere and Abril (2012) and Venkiteswaran et al. (2014). They are independent of the gas transfer velocity $k$ and account for the upstream portion of a sampling site in headwater streams.

The aim of this work was to model stream $CO_2$ outgassing on the basis of the stable isotope approach by Polsenaere and Abril (2012) and to extend their method by including measured groundwater stable isotope signatures in order to reduce modelling uncertainties. Our study utilizes data from the well-studied Uhlirska catchment in the Jizera Mountains (Czech Republic) (Dusek et al., 2012;Sanda et al., 2014;Vitvar et al., 2016;Marx et al., 2017b). Since the background geology of the catchment consists of silicate rocks, carbonate weathering as $CO_2$ source can be virtually excluded. The $CO_2$ saturation is then exclusively controlled by the mobilization of terrestrial respired organic carbon and by the input of shallow groundwater (Humborg et al., 2010;Amiotte-Suchet et al., 1999).

## 2 Materials and methods

### 2.1 Study site

The Uhlirska catchment is situated in the northern Czech Republic, 9 km northeast of the city of Liberec (Fig. 1). The stream Cerna Nisa flows in the catchment valley and is a tributary of the Luzicka Nisa River that later merges with the Odra River and flows towards the Baltic Sea. The stream length in this experimental catchment is about 2100 meters and water travel times are less than one hour from the spring to the catchment outlet.

Table 1 lists the main characteristics of the Uhlirska catchment. The annual average precipitation exceeds 1200 mm yr$^{-1}$ and the annual average temperature is 5.5 °C (1996-2009). Snowcover typically prevails during 6 months of the year, mostly between November and March (Hrncir et al., 2010). During our study period from September 2014 to April 2016 snowcover prevailed only between January and March. However, during snowmelt the monthly average discharge doubles (>50 L s$^{-1}$) compared to the other months' discharges (Table 1).

The forest consists of a spruce monoculture (*Picea abies*) and isolated patches of larch, beech and rowan trees (Sanda and Cislerova, 2009). Purely granitic bedrocks underlie this type of C3 vegetation. The catchment has two basic types of soils. On the hillslopes, about 60-90 cm deep and highly heterogeneous soil profiles consist of dystric Cambisols, Podzols or Cryptopodzols that developed on weathered and fractured bedrocks. These soil types cover approximately 90 % of the catchment area. The valley bottom soils consist of a layer of peat of mostly Histosol-types with depths up to 300 cm. The latter soils cover approximately 10 % of the catchment area, make up small wetlands along the stream and lie on top of fluvial material, which embodies the main perennial aquifer (Sanda et al., 2014).

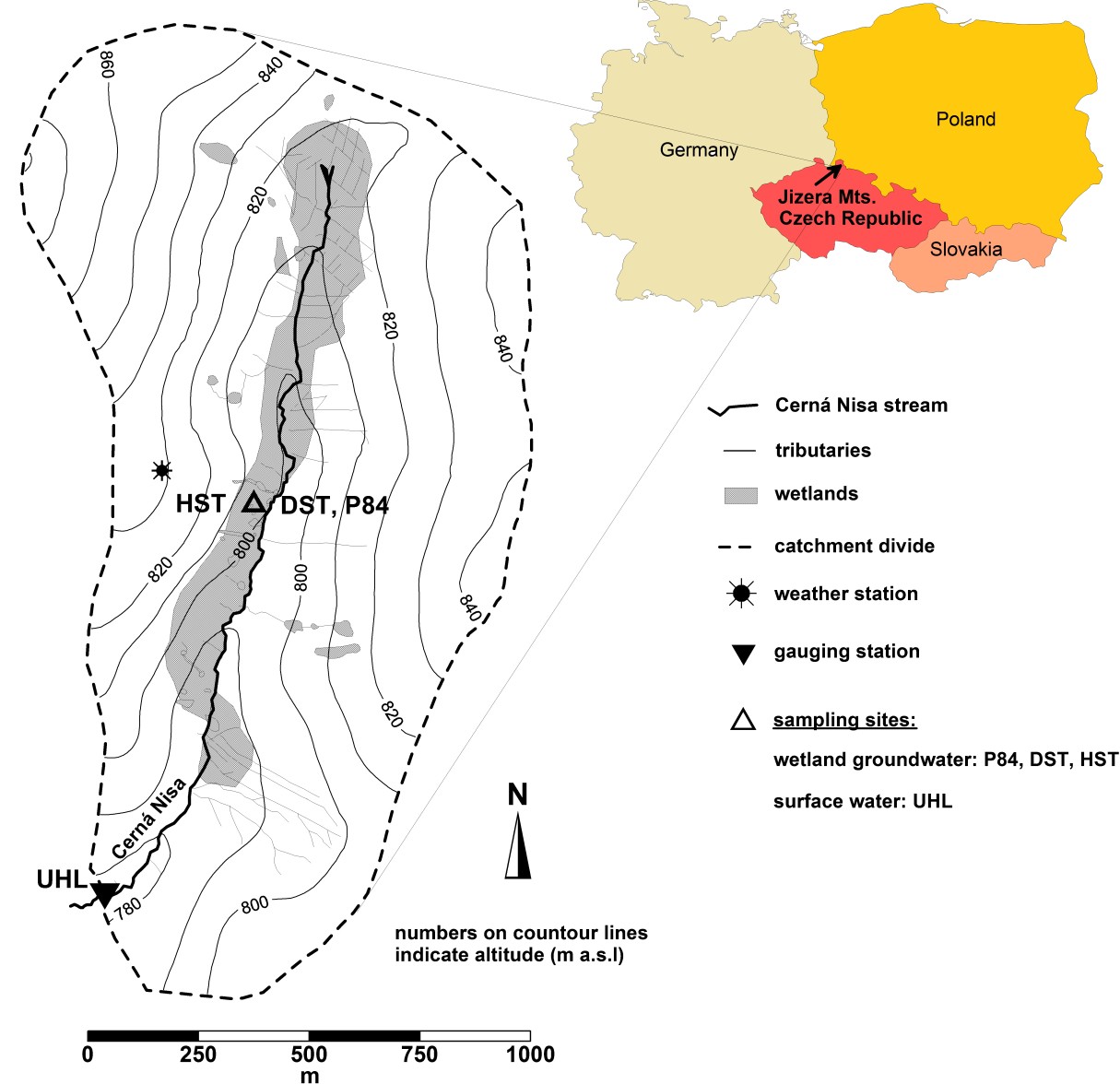

**Figure 1: Location of the Uhlirska catchment and sampling sites modified from Sanda et al. (2014).**

**Table 1: Characteristics of the Uhlirska catchment and Cerna Nisa stream.**

| | |
|---|---|
| Location | 15°09'E, 50°49'N |
| Altitude range | 776-886 m above sea level |
| Drainage area | 1.78 km$^2$ |
| Stream surface area | 1490 m$^2$ |
| Strahler stream order | 1 |

| Total stream length | 2.1 km |
|---|---|
| Stream flow (Q) | $Q_{median}$= 21.2 L s$^{-1}$ (2014-2016) |
| | Snowmelts in 2015 and 2016: Q> 50 L s$^{-1}$ |
| Mean slope | 2.3 % |
| Annual average air temperature | 5.5 °C (1996-2009) |
| Annual average precipitation | 1212 mm (1996-2009) |
| Dominant vegetation | 95 % Norway spruce, 5 % grassland |
| Dominant soil type | Cambisols, Podzols, Cryptopodzols, Histosols, Gleysols |
| Bedrock | Granite, deluviofluvial sediments, glacial tills |

## 2.2 Water sampling and laboratory analyses

Between September 2014 and April 2016 shallow wetland groundwater was collected on a monthly and stream water on approximately a weekly basis. Sampling points are shown in Figure 1.

The stream discharge was determined at a V-notch weir at the catchment outlet (site UHL, see Fig. 1). Temperature (T), pH
and total alkalinity (TA) were determined in the field with portable HACH equipment that included a multi-parameter instrument and a digital titrator (all HACH Company, Loveland, CO, USA). The measured parameters exhibited a precision of 0.1 pH units, 0.1 °C (2σ) for T and was better than 2 % (2σ) per 100 titration steps for TA (Marx et al., 2017b). Water samples were collected from the stream approximately 10 cm below the water surface and from boreholes with a peristaltic pump approximately 24 hours after purging the boreholes.

Particulate organic carbon (POC) was sampled on 0.7 $\mu$m pore size glass fibre filter papers and analysed according to Barth et al. (2017). All other samples were filtered via syringe disk filters with 0.45 $\mu$m pore size (Minisart HighFlow PES, Sartorius AG, Germany) in the field. Before filtration, both syringe and membrane were pre-washed with sample water. Samples for the analysis of dissolved inorganic carbon (DIC) concentrations and isotopes were collected in 40 mL amber glass vials without headspace. The vials fulfil specifications of the US Environmental Protection Agency (EPA) and were
closed with butyl rubber/PTFE septa and open-hole caps, with butyl rubber side showing towards the sample. Vials were poisoned with 20 $\mu$L of a saturated HgCl$_2$ solution to avoid biological activity after sampling. After collection, all samples were kept in the dark at 4°C until analyses.

Water samples were analysed for their carbon stable isotope ratios of dissolved inorganic carbon ($\delta^{13}C_{DIC}$) and dissolved organic carbon ($\delta^{13}C_{DOC}$) by an OI Analytical Aurora 1030W TIC-TOC analyser (OI Analytical, College Station, Texas) that
was coupled in continuous flow mode to a Thermo Scientific Delta V plus isotope ratio mass spectrometer (IRMS) (Thermo Fisher Scientific, Bremen, Germany). The sample was reacted with 1 mL of 5 % phosphoric acid (H$_3$PO$_4$) at 70 °C for 2 min to release the dissolved inorganic carbon (DIC) as CO$_2$. The evolved CO$_2$ was purged from the sample by helium. In a second step 2 mL of 10 % sodium persulfate (Na$_2$S$_2$O$_8$) were reacted for 5 min at 98 °C to oxidize the DOC to CO$_2$ that was

subsequently purged from the solution by helium. A trap and purge (T&P) system was installed for the analysis of low concentrations. Details of the coupling of the TIC/TOC analyser to IRMS are described in St-Jean (2003).

All values are reported in the standard $\delta$-notation in per mil (‰) vs. Vienna Pee Dee Belemnite (VPDB) according to

$$\delta = \frac{R_{\text{sample}}}{R_{\text{reference}}} - 1 ,$$
(1)

where $R$ is the ratio of the numbers ($n$) of the heavy and light isotope of an element (e.g. $n(^{13}C)/n(^{12}C)$) in the sample and the reference (Coplen, 2011). The data sets were corrected for instrumental drift during the run and linearity. The data were normalized to the VPDB scale by two laboratory reference materials ($C_4$ sugar and KHP) measured in each run. The in-house reference materials were calibrated directly against USGS-40 (L-glutamic acid) and IAEA-CH-6 (sucrose) by using an elemental analyser (Costech ECS 4010). A value of –26.39 ‰ and –10.45 ‰ was assigned to USGS-40 and IAEA-CH-6, respectively. Concentration was determined from the signal of the OI Aurora 1030W internal non-dispersive infrared sensor (NDIR) and a set of calibration standard with known concentrations prepared from analytical (A.C.S.) grade potassium hydrogen phthalate (KHP). Areas of the sample peaks are directly proportional to the amount of $CO_2$ generated by the reaction of the sample with acid (DIC) or sodium persulfate (DOC). Analytical precision based on the repeated analyses of a control standard ($C_3$ sugar) during all runs was better than ±0.3 ‰ for $\delta^{13}C$ and better than 5% for concentrations (1σ).

POC samples were analysed for $\delta^{13}C_{\text{POC}}$ using a Costech Elemental Analyser (ECS 4010; Costech International, Pioltello, Italy; now NC Technologies, Bussero, Italy) coupled in continuous flow mode to a Thermo Scientific Delta V plus IRMS. The data sets were corrected for linearity and instrumental drift during the run. Values were normalized for carbon to VPDB by analyses of internal reference materials ($C_4$ sugar and KHP) that were calibrated directly versus USGS-40 and USGS-41 (L-glutamic acid). Assigned values to USGS-40 and USGS-41 were –26.39 ‰ and +37.63 ‰ for $\delta^{13}C$, respectively. For precision and accuracy two laboratory standards (acetanilide and tartaric acid) were measured in each run. Precision, defined as the standard deviation of the control standard was better than 0.1 ‰ (1σ) for $\delta^{13}C_{\text{POC}}$.

### 2.3 Calculations and model assumptions

At the Uhlirska headwater catchment the original streamCO$_2$-DEGAS model was applied to calculate stream $CO_2$ outgassing for different scenarios with varying values of river respiration (R) with 10, 19, 25, 50, and 75 % to test the model sensitivity to these values. In a second approach we modified the streamCO$_2$-DEGAS model as follows: instead of soil organic matter ($\delta^{13}C_{\text{SOM}}$) we used groundwater $\delta^{13}C_{\text{DIC}}$ data to better constrain initial $CO_2$ values and to reduce the model uncertainty.

The Uhlirska catchment meets the assumption of the streamCO$_2$-DEGAS model with (i) stream waters being acidic with pH values between 3.9 and 6.7 (Table 2), and for our study we assumed that (ii) waters in the stream are unproductive. This means that secondary processes such as photosynthesis by algae or biofilms and DOC degradation to $CO_2$ are neclected. This is a plausible assumption because high runoff and short residence times often leave insufficient time for substantial

degradation of DOC (Raymond et al., 2016;Catalan et al., 2016). However, the potential of temperature-dependent in-stream bio- and photodegradation (Demars et al., 2011;Moran and Zepp, 1997), particularly during summer cannot be entirely excluded in the Cerna Nisa stream.

The original streamCO$_2$-DEGAS model by Polsenaere and Abril (2012) demands the input variables of total alkalinity (TA), $\delta^{13}C_{DIC}$, $pCO_2$ and temperature (Table 2) as well as the proportion of river respiration ($R$) and the carbon isotope composition of $\delta^{13}C_{SOM}$. Note that TA and groundwater $\delta^{13}C_{DIC}$ were only measured during monthly samplings and linear interpolated otherwise.

We used DIC concentrations to calculate $HCO_3^-$ (Dickson, 2007) and together with pH and T we were able to calculate $pCO_2$ values with the following equation (Plummer and Busenberg, 1982):

$$pCO_2 = \frac{HCO_3^- \times H^+}{K_H \times K_1}, \tag{2}$$

where $HCO_3^-$ is the activity of bicarbonate, $H^+$ is $10^{-pH}$, $K_1$ is the temperature-dependent first dissociation constant for the dissociation of $H_2CO_3$ (all variables in mol $L^{-1}$), and $K_H$ is the Henry's law constant in mol $L^{-1}$ atm$^{-1}$. The uncertainty of the calculation depends on the measurement uncertainties of DIC, pH and T. The largest uncertainty is caused by pH measurements as they are on a logarithmic scale. The pH measurement uncertainty is typically smaller than ±0.1 pH units and causes a maximum uncertainty of ±21 % for $pCO_2$. We consider this as a worst-case scenario that is also indicated in Figure 3.

R (0 < R < 1) is the proportion of $CO_2$ resulting from respiration in water along the entire stream course (waterlogged soils, stream waters, and sediments) and was approximated by a range between 0.1 and 0.19 (i.e., 10 to 19 %). This corresponds to the credible range of internal $CO_2$ production as a percentage of median stream $CO_2$ emissions from small streams (<0.01 m$^3$ s$^{-1}$) in the contiguous United States and was determined by Hotchkiss et al. (2015).

POC was considered as best representative of soil organic matter in the catchment. It is the material closest to the original plant and soil material. DOC has recalcitrant phases and thus would introduce a non-representative value for organic matter input (Laudon et al., 2011;Cauwet and Sidorov, 1996). Polsenaere and Abril (2012) chose a $\delta^{13}C_{SOM}$ of −28 ‰ that was determined as average annual $\delta^{13}C_{POC}$ at their study site. In our study we also used the average annual stream $\delta^{13}C_{POC}$. Thus the $\delta^{13}C_{SOM}$ was confined with −29.5 ‰ (Table 2). In addition, the average of wetland groundwater $\delta^{13}C_{DIC}$ with an assumed isotopic fractionation of +1 ‰ for movement of $CO_2$ in waterlogged soils, groundwaters, river waters and sediments (O'Leary, 1984), served as input of initial $\delta^{13}C_{DIC}$ ($\delta^{13}C$-DIC$_{init.}$). Measured $\delta^{13}C_{DIC}$ values ranged between −25.2 ‰ and −10.1 ‰ (Table S1).

The model results are the partial pressure of initial $CO_2$ before gas exchanges ($pCO_{2init}$) with the atmosphere start and the fraction of stream DIC that has degassed into the atmosphere ([DIC]$_{ex}$). [DIC]$_{ex}$ corresponds to the $CO_2$ loss upstream of the sampling point and is given as a concentration (Polsenaere and Abril, 2012). $CO_2$ fluxes were calculated by multiplication

with average discharges that were established from daily values. To allow for inter-catchment comparisons the carbon losses were normalised to the stream surface area. In addition, to avoid often imprecise stream lengths and surface areas, the carbon losses were also normalised to the catchment area.

For comparison, the gas transfer velocity adjusted to the in situ temperature ($k_T$, in m d$^{-1}$) can be calculated from model results when assuming that the water ($pCO_{2,aq}$) is the average between the modelled soil $pCO_2$ and the in situ $pCO_2$ at the catchment outlet:

$$k_T = \frac{F}{K_H \times \left( pCO_{2,aq} - pCO_{2,air} \right) \times M_C} \, , \tag{3}$$

where F is the modelled $CO_2$ outgassing (in g m$^{-2}$ d$^{-1}$), $pCO_{2,air}$ the partial pressure of $CO_2$ in the atmosphere considered with ~400 ppmV (ESRL/GMD, 2017), $K_H$ is the Henry's law constant (in mol L$^{-1}$ atm$^{-1}$) and $M_C$ the molar mass of C (12.011 g mol$^{-1}$). $k_T$ was then converted into the normalized gas transfer velocity of $CO_2$ at 20°C ($k_{600}$, in m d$^{-1}$) according to:

$$k_{600} = \frac{k_T}{\left( \frac{Sc_T}{600} \right)^{-0.75}} \, , \tag{4}$$

where $Sc_T$ is the Schmidt number at the measured in situ temperature (Raymond et al., 2012).

All modelling approaches were executed via Matlab (MathWorks, Natick, MA, USA).

**2.4 Model input variables from the Uhlirska catchment**

Stream (UHL) and wetland groundwater (DST, HST, P84) pH values were always below 6.7 during the entire study period (Table 2). In agreement with generally low stream pH values, the capacity of waters to buffer acidic inputs indicated by TA was generally low. The TA ranged around a median of 130 $\mu$mol L$^{-1}$ with a standard deviation (1$\sigma$) of ±101 $\mu$mol L$^{-1}$ at the catchment outlet (Table 2). Note, that in the following the standard deviations express the variation over the sampling period and not the measurement error. The $\delta^{13}C_{DIC}$ values had a median of –18.4 ‰ with a range from –25.2 to –10.1 ‰ (Table 2). The most negative values were measured in wetland groundwater with median values of –24.5 ‰, –28.7 ‰ and –24.3 ‰ in HST, DST, and P84, respectively (Fig. 1). The range of these groundwaters between –23.6 and –29.6 ‰ fits the range of $\delta^{13}C$ in C3 plants and associated soil organic matter. In the different water compartments (wetland groundwater and stream, Table 2), the DIC concentrations decreased with increasing $\delta^{13}C_{DIC}$ values. The lowest DIC concentrations were found at the catchment outlet (31 to 483 $\mu$mol L$^{-1}$) and the highest concentrations in wetland groundwater (284 to 540 $\mu$mol L$^{-1}$). The median stream $pCO_2$ determined at the catchment outlet was 1374 ±710 ppmV with a range from 450 to 3749 ppmV. Values in wetland groundwater were typically higher with median values of 5590, 4440, and 6210 ppmV at HST, DST, and P84, respectively. Stream water temperatures showed a characteristic annual trend with a range from 0.8 °C to 13.8 °C and a median of 4.5 °C (Table 2).

**Table 2: pH, total alkalinity (TA), temperature (T), $p$CO$_2$ values, DIC concentrations and $^{13}$C/$^{12}$C ratios expressed as median values with ± 1σ standard deviation and ranges given in parentheses. Site names are according to Figure 1.**

| Site ID | Description [sampling depth] | pH | TA $\mu$mol L$^{-1}$ | T °C | DIC $\mu$mol L$^{-1}$ | $\delta^{13}$C$_{DIC}$ ‰ | $p$CO$_2$ ppmV | $\delta^{13}$C$_{POC}$ ‰ |
|---|---|---|---|---|---|---|---|---|
| | *Surface water* | | | | | | | |
| UHL | Main stream outlet | 5.9 ±0.7 | 130 ±101 | 4.5 ±3.5 | 114±72 | –18.4±3.3 | 1374 ±710 | –29.5 ±0.6 |
| | | (3.9/6.7) | (24/500) | (0.8/13.8) | (31/483) | (–25.2/–10.1) | (450/3749) | (–30.5/–27.5) |
| | *Groundwater* | | | | | | | |
| HST | Wetland | 5.9 ±0.2 | 180 ±57 | 6.4 ±2.1 | 424 ±46 | –24.5 ±0.8 | 5590 ±1120 | –28.6 ±1.0 |
| | [2.7 m] | (5.7/6.4) | (120/350) | (4.3/11.3) | (349/509) | (–27.0/–23.6) | (3660/7860) | (–29.8/–25.6) |
| DST | Wetland | 5.9 ±0.4 | 160 ±38 | 5.9 ±2.1 | 399 ±57 | –28.7 ±0.7 | 4440 ±1410 | –28.7 ±0.7 |
| | [3.7 m] | (5.7/7.3) | (82/244) | (3.1/10.8) | (284/482) | (–29.8/–27.6) | (1080/6960) | (–29.8/–27.6) |
| P84 | Wetland | 5.9 ±0.5 | 158 ±33 | 6.8 ±1.4 | 499 ±58 | –24.3 ±0.7 | 6210 ±1720 | –29.2 ±0.6 |
| | [5.2 m] | (5.0/7.3) | (110/222) | (3.3/9.6) | (299/540) | (–25.0/–22.1) | (980/7830) | (–29.9/–28.1) |

## 3 Results and discussion

Our study shows that the uncertainty of the respiration parameter ($R$) can be circumvented by incorporating wetland groundwater $\delta^{13}$C$_{DIC}$ into the streamCO$_2$-DEGAS model. In a first step the original CO$_2$-DEGAS model was run with an $R$ value of 10, 19, 25, 50, and 75 %. In a second step fractionation-corrected groundwater $\delta^{13}$C$_{DIC}$ was incorporated into the model. Thus the isotope composition of the initial CO$_2$ was better constrained and the uncertainty on the isotope fractionation in soils was reduced.

In addition, we were able to estimate DIC export with modelled CO$_2$ outgassing and calculated lateral export of HCO$_3^-$, CO$_2$* as well as of total DIC (Table 3). These data cover a period of 20 months and measurements took place at the catchment outlet, whereas modelling results relate to CO$_2$ outgassing between the stream source and the catchment outlet (UHL).

### 3.1 Model sensitivity

The parameter of proportion river respiration ($R$) has a large impact on the results by the model. It is attributed to the average percentage of respiration occurring along the entire stream course (waterlogged riparian soils, stream waters, sediments and the hyporheic zone). Thus, it is important to evaluate the model sensitivity to assumed respiration. Modelled CO$_2$ outgassing and modelled initial $p$CO$_2$ values relate to an assumed $R$ of 10 and 19 %, which corresponds to an average in-stream respiration as determined by Hotchkiss et al. (2015). However, this value does not account for respiration in groundwater and soil water. The contributions of these compartments are typically also considerable in headwaters. Although this type of respiration was not measured directly, we can assume a large potential of respiration in waters of the organic-rich wetland and of riparian soils with peak values during late summer and early fall (Pacific et al., 2008). In addition, respiration in

gravel bar waters along the stream (Boodoo et al., 2017) can lead to an exceedance of 19 % for R along the Cerna Nisa stream.

In some cases the model was not able to produce reliable results and the convergence criteria were not fulfilled. The initial soil $p$CO$_2$ had to be extremely large (> 150000 ppmV) for selected sampling events to reach the convergence of both $\delta^{13}$C$_{DIC}$ and $p$CO$_2$ at the same iteration (Polsenaere and Abril, 2012). This was during few cases in February and March 2015 and during summer between May and September 2015. However, for an $R$ of 10 to 19 % during 22 to 23 cases, for an $R$ of 25 and 50 % during 26 cases and for an $R$ of 75 % during 29 cases the convergence criteria were not attained. Consequently, with increasing $R$ a necessary convergence was more often not possible due to exceeding reasonable boundaries. These findings suggest that 10 to 19 % are reasonable $R$ values for most months, except for June to August in 2015. Low respiration values are plausible for springtime, however during summer an increased respiration can be assumed due to warmer temperatures and increased biological activity. One possible explanation for higher model uncertainty during summer is that the modelled CO$_2$ loss shows a strong non-linear dependence on in situ $p$CO$_2$ and $\delta^{13}$C$_{DIC}$ (Polsenaere and Abril, 2012). Thus, the relative error of modelled CO$_2$ loss increases with the modelled CO$_2$ loss itself. This is particularly the case for low TA (e.g. 0.1 mmol L$^{-1}$ (Polsenaere and Abril, 2012)), where the $\delta^{13}$C$_{DIC}$ increase is not buffered by the HCO$_3^-$ pool. According to Polsenaere and Abril (2012), large losses of CO$_2$ only occur with high in situ $p$CO$_2$ and with high $\delta^{13}$C$_{DIC}$. TA was low with 218 and 196 $\mu$mol L$^{-1}$ in the Uhlirska catchment during June and July 2015. In contrast, $\delta^{13}$C$_{DIC}$ values and in situ $p$CO$_2$ were increased with on average –15.8 and –16.2 ‰ as well as 2225 and 2411 ppmV, respectively. Thus, increased $\delta^{13}$C$_{DIC}$ values together with the model's non-linear dependence on in situ $p$CO$_2$ and $\delta^{13}$C$_{DIC}$ may have caused an overestimation of modelled CO$_2$ losses and – as a consequence – led to failed convergence criteria during summer. Note that processes such as photosynthesis, which are not part of the model, may also lead to increased $\delta^{13}$C$_{DIC}$ and lower in situ $p$CO$_2$ values. This may further increase the uncertainty of model results.

## 3.2 Gas transfer velocities

A huge benefit of the applied isotope approach is, that the difficult to estimate gas transfer velocity parameter ($k$) is not needed for the calculation of CO$_2$ outgassing fluxes. On the other hand, $k_{600}$ (gas transfer velocity of CO$_2$ at 20°C, eq. (4)) values could be estimated from model results when assuming the water $p$CO$_2$ being the average between the modelled soil $p$CO$_2$ and the in situ $p$CO$_2$ at the catchment outlet (Table 3; Figure 2).

For the original model calculated $k_{600}$ values varied from 0.5 to 7.0 m d$^{-1}$ and from 0.6 to 7.9 m d$^{-1}$ for $R = 10$ and 19 % (Table 3), with higher values mostly occurring during springtime and lower values during summer (Fig. 2). These values were higher than $k_{600}$ values that were determined in a lowland boreal landscape of Québec (Campeau et al., 2014)(Table 4). However, values were slightly higher than in the silicate Renet catchment in France (1.2 – 7.2 m d$^{-1}$) and were comparable to results from temperate silicate catchments in Germany (Halbedel and Koschorreck, 2013) (Table 4). They also equalled median values for boreal and arctic streams (latitude 50°-90°) determined by Aufdenkampe et al. (2011), whereas $k_{600}$ values

for Sweden, for the United States, and for this study - all determined with model equations (i.e., Raymond et al. (2012)) - showed higher values (Table 4). For modelling with groundwater data, we obtained lower $k_{600}$ values than with the original model (Table 3). Values ranged from 0.1 to 5.4 m d$^{-1}$ (Table 3) with higher values during springtime and lower values during summer (Fig. 2). The mean $k_{600}$ value was similar to the value of 2.2 m d$^{-1}$ calculated from gas transfer coefficients for propane in the temperate Zilllierbach stream in Germany (Table 4).

**Table 3: DIC partitioning according to the streamCO$_2$-DEGAS model (Polsenaere and Abril, 2012) and calculated from DIC measurements from September 2014 to April 2016 in the Uhlirska catchment expressed as median/average values ± 1σ standard deviation and ranges given in parentheses. The total DIC export corresponds to the sum of HCO$_3^-$ export, CO$_2$* export and CO$_2$ outgassing. Initial soil $p$CO$_2$ was calculated from DIC$_{init.}$, pH and temperature. $k_{600}$ was calculated from model results.**

| | |
|---|---|
| *Concentrations* | |
| HCO$_3^-$ from silicate weathering (mg L$^{-1}$) | 0.29/0.49 ±0.56 (0.001 – 3.44) |
| CO$_2$* dissolved in water (mg L$^{-1}$) | 1.04/1.09 ±0.44 (0.37 – 2.36) |
| CO$_2$ loss [a] (mg L$^{-1}$) | 1.93/6.35 ±12.43 (0.01 – 72.52) |
| CO$_2$ loss [b] (mg L$^{-1}$) | 1.34/7.47 ±14.99 (0.003 – 85.40) |
| Total DIC exported from land [a] (mg L$^{-1}$) | 3.72/7.93 ±12.72 (0.38 – 74.64) |
| Total DIC exported from land [b] (mg L$^{-1}$) | 2.88/9.04 ±15.37 (0.38 – 87.73) |
| Modelled soil $p$CO$_2$ [a] ($p$CO$_{2init.}$) (ppmV) | 2790/7670 ±16490 (460 – 106770) |
| Modelled soil $p$CO$_2$ [b] ($p$CO$_{2init.}$) (ppmV) | 2730/10360 ±22910 (460 – 131050) |
| | |
| *Fluxes (normalised to the catchment area)* | |
| HCO$_3^-$ from silicate weathering (mg C m$^{-2}$ d$^{-1}$) | 0.36/0.40 ±0.35 (0.02 – 1.63) |
| CO$_2$* dissolved in water (mg C m$^{-2}$ d$^{-1}$) | 1.27/1.91 ±1.89 (0.26 – 11.68) |
| CO$_2$ outgassed to the atmosphere [a] (mg C m$^{-2}$ d$^{-1}$) | 1.92/5.10 ±8.37 (0.03 – 57.07) |
| CO$_2$ outgassed to the atmosphere [b] (mg C m$^{-2}$ d$^{-1}$) | 1.66/4.93 ±8.42 (0.02 – 49.74) |
| Total DIC exported from land [a] (mg C m$^{-2}$ d$^{-1}$) | 4.89/7.42 ±8.26 (1.38 – 58.61) |
| Total DIC exported from land [b] (mg C m$^{-2}$ d$^{-1}$) | 4.04/7.24 ±8.18 (1.28 – 51.10) |
| | |
| *Fluxes (normalised to the stream surface area)* | |
| HCO$_3^-$ from silicate weathering (g C m$^{-2}$ d$^{-1}$) | 0.43/0.48 ±0.42 (0.03 – 1.95) |
| CO$_2$* dissolved in water (g C m$^{-2}$ d$^{-1}$) | 1.53/2.29 ±2.26 (0.31 – 13.98) |
| CO$_2$ outgassed to the atmosphere [a] (g C m$^{-2}$ d$^{-1}$) | 2.29/6.10 ±10.02 (0.03 – 68.29) |
| CO$_2$ outgassed to the atmosphere [b] (g C m$^{-2}$ d$^{-1}$) | 1.99/5.90 ±10.08 (0.02 – 59.52) |
| Total DIC exported from land [a] (g C m$^{-2}$ d$^{-1}$) | 5.85/8.87 ±9.88 (1.65 – 70.13) |
| Total DIC exported from land [b] (g C m$^{-2}$ d$^{-1}$) | 4.83/8.67 ±9.79 (1.54 – 61.15) |
| | |
| $k_{600}$ [a] (m d$^{-1}$) | 2.8/3.1 ±1.4 (0.5 – 7.9) |
| $k_{600}$ [b] (m d$^{-1}$) | 1.9/2.1 ±1.0 (0.1 – 5.4) |

[a] Modelling with proportion river respiration (R) = 10 and 19 %.
[b] Modelling with groundwater data.

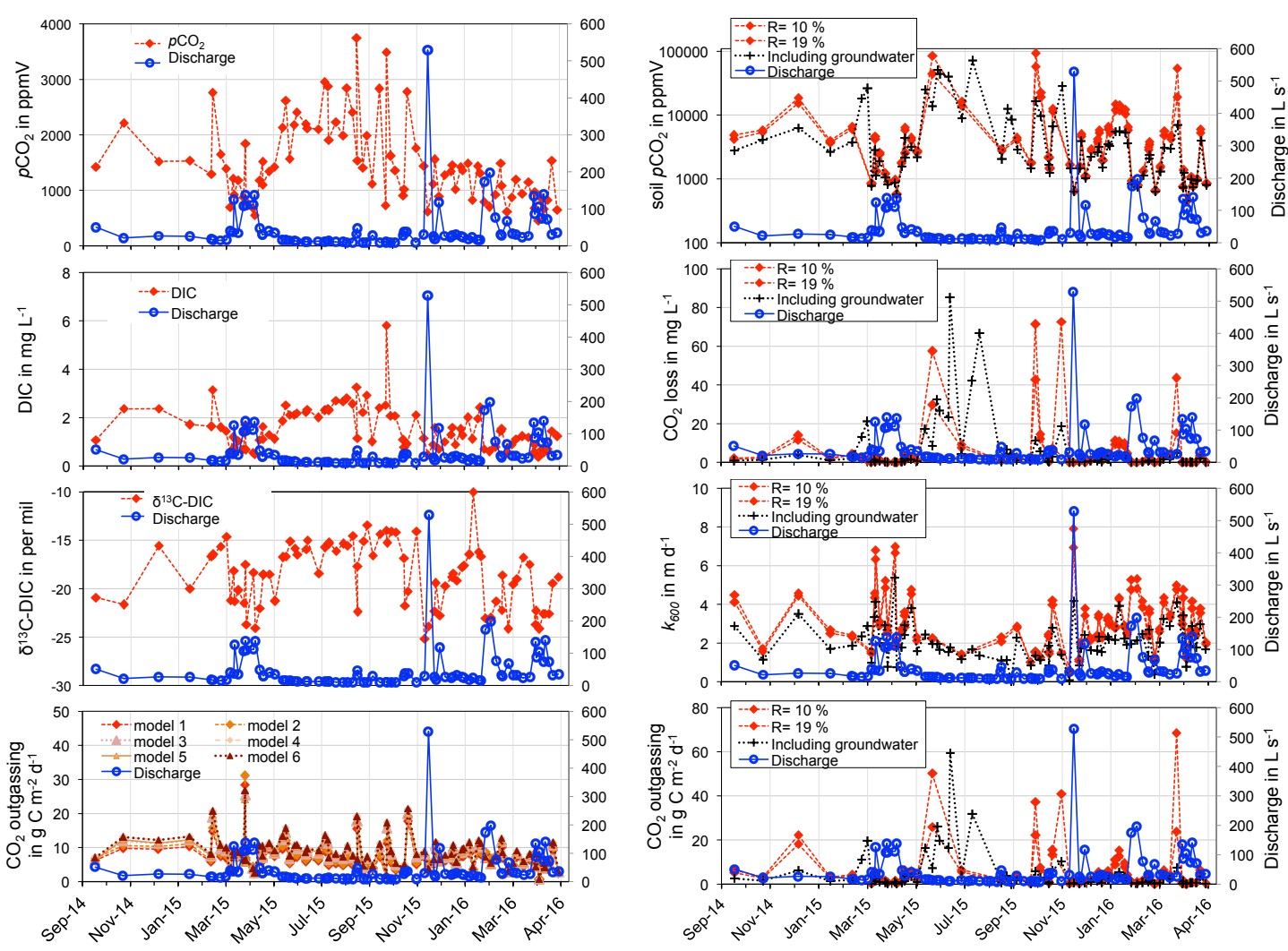

**Figure 2: Data of measured discharge, in situ $p$CO$_2$, $\delta^{13}$C$_{DIC}$, DIC concentrations, and CO$_2$ fluxes calculated via model equations in Raymond et al. (2012) for the Uhlirska catchment outlet (UHL) (left) and modelled soil $p$CO$_2$, CO$_2$ loss as well as from model results calculated $k_{600}$ and CO$_2$ outgassing normalized to the stream surface area (right).**

5 **Table 4. Gas transfer velocities ($k_{600}$) normalized to a stream temperature of 20°C of low order streams. Values calculated from model results are displayed in Table 3.**

| Region | Stream order | $k_{600}$ in m d$^{-1}$ | Reference |
|---|---|---|---|
| Temperate streams (25°-50°) | - | 4.8 [b, c] | Aufdenkampe et al. (2011) |
| Uhlirska, Czech Republic | 1 | 6.4 [a] 6.0 [b] | This study (model 1 – 6, Raymond et al. (2012) |
| Renet, France | - | 2.9 [a] | Polsenaere and Abril (2012) |
| United States | <4 | 4.5 [b] | Butman and Raymond (2011) |
| Wiesent, Germany | - | 6.3 [a] | van Geldern et al. (2015) |
| Rappbode, Germany | - | 2.9 [a] | Halbedel and Koschorreck (2013) |

| | | | |
|---|---|---|---|
| Hassel, Germany | - | 2.4 [a] | Halbedel and Koschorreck (2013) |
| Zillierbach, Germany | - | 2.2 [a] | Halbedel and Koschorreck (2013) |
| Ochsenbach, Germany | - | 2.5 [a] | Halbedel and Koschorreck (2013) |
| Boreal and arctic streams (50°-90°) | - | 3.1 [b, c] | Aufdenkampe et al. (2011) |
| Québec, Canada | 1 | 0.6 [a] | Campeau et al. (2014) |
| Québec, Canada | 2 | 0.6 [a] | Campeau et al. (2014) |
| Québec, Canada | 3 | 0.5 [a] | Campeau et al. (2014) |
| Québec, Canada | 4 | 1.4 [a] | Campeau et al. (2014) |
| Alaska, United States | ≦4 | 6.5 [a] | Crawford et al. (2013) |
| Sweden | 1 | 15.5 [a] | Humborg et al. (2010) |
| Sweden | 2 | 12.4 [a] | Humborg et al. (2010) |

[a] Mean values.

[b] Median values.

[c] Running waters in Aufdenkampe et al. (2011) have $< 60 - 100$ m width.

### 3.3 Soil $pCO_2$

The streamCO2-DEGAS model assumes that the modelled initial $pCO_2$ represents soil $pCO_2$ (Polsenaere and Abril, 2012). For the Uhlirska catchment, this would mean that soil $pCO_2$ values ranged between 460 and 106770 ppmV for original model and between 460 and 131050 ppmV for modelling with groundwater input (Table 3). These were mostly within the range of values reported by Amundson and Davidson (1990) who determined $pCO_2$ from 400 to 130000 ppmV for the upper metre of soils around the world. For three acidic catchments in France, modelled soil $pCO_2$ varied between 2120 and 77860 ppmV (Polsenaere and Abril, 2012). In a temperate hardwood-forested catchment Jones and Mulholland (1998) modelled soil $pCO_2$ values between 907 in winter and 35313 ppmV in summer. Higher soil $pCO_2$ values were also modelled during summer in the Uhlirska catchment. The main reasons for elevated soil $pCO_2$ in warmer seasons are higher temperatures with enhanced soil respiration in summer (Jones and Mulholland, 1998).

A clear positive correlation between soil $pCO_2$ and $CO_2$ outgassing for modelling with groundwater inputs ($r^2 > 0.96$, Table S2) stresses the importance of soil $pCO_2$ values and their dynamics for stream $CO_2$ loss.

### 3.4 CO2 outgassing

Figure 3A displays modelled $CO_2$ outgassing for the proportion of river respiration ($R$) inputs that were selected with 10, 19, 25, 50, and 75 %. Figure 3B displays results for the modified modelling with groundwater input. Shaded areas show $pCO_2$ measurement uncertainties. Points that plot outside the measurement uncertainty did not fulfil the convergence criteria (Polsenaere and Abril, 2012). They were thus replaced by interpolated values in the discussion. Note that, for modelling with groundwater data the fluxes where the convergence criteria were not fulfilled could get reduced by ~50 % (Fig. 3B).

For modelling with $R$ = 10 % and 19 % the modelled $CO_2$ outgassing varied from 0.01 to 72.52 mg C $L^{-1}$ and showed a mean of 6.35 mg C $L^{-1}$ (Table 3). When normalised to the catchment area, fluxes showed a mean of 5.10 mg C $m^{-2}$ $d^{-1}$ with values from 0.03 to 57.07 mg C $m^{-2}$ $d^{-1}$.

The highest fluxes > 30 mg C $m^{-2}$ $d^{-1}$ were modelled in May 2015, September 2015, November 2015, and March 2016 (Fig. 2). Translating these results to annual carbon outgassing fluxes yielded a mean of 23.9 to 34.5 g C $m^{-2}$ $yr^{-1}$ for 12 consecutive months and $R$ = 10 and 19 %. Normalised to the stream surface area, an average of 28.6 to 41.3 kg C $m^{-2}$ $yr^{-1}$ were outgassed annually.

When including shallow wetland groundwater data in the model, the mean $CO_2$ outgassing was 1.34 mg C $L^{-1}$ and varied from 0.003 to 85.40 mg C $L^{-1}$, with minimum and maximum values during April 2016 and June 2015 (Table 3). Corresponding fluxes with a mean of 8.8 kg C $d^{-1}$ and a maximum of 88.5 kg C $d^{-1}$ were smaller when compared to mean $CO_2$ losses of 952, 258, and 10671 kg C $d^{-1}$ for similar organic-rich catchments in France calculated with the same model approach (Polsenaere and Abril, 2012). However, stream discharge in the French catchments was larger by at least one order of magnitude, thus yielding higher outgassing rates. When normalised to the catchment area, modelled $CO_2$ losses had a mean of 4.9 mg C $m^{-2}$ $d^{-1}$ and varied from 0.02 to 49.7 mg C $m^{-2}$ $d^{-1}$ in the Uhlirska catchment. These values translate to a mean 5.9 g C $m^{-2}$ $d^{-1}$ and a range between 0.02 and 59.5 g C $m^{-2}$ $d^{-1}$, when normalised to the stream surface area. Those groundwater-improved $CO_2$ outgassing estimates yielded the same outgassing trend with slightly decreased values (Fig. 3).

The most used method to calculated $CO_2$ fluxes is via $k$ values, which are predicted via slope, flow velocity, stream depth, and discharge (Raymond et al., 2012). Corresponding equations (1) to (6) yielded a mean flux of 8.1 g C $m^{-2}$ $d^{-1}$ with a range between 0.6 and 31.1 g C $m^{-2}$ $d^{-1}$ for the Uhlirska catchment. This mean value was larger than model results, whereas the range of fluxes was smaller (Table 3; Figure 2). Huotari et al. (2013) and Wallin et al. (2011) observed large uncertainties of calculated $k$ values and corresponding outgassing fluxes on the temporal scale. For instance, in headwater streams Wallin et al. (2011) determined errors of up to 100 % in median outgassing rates compared to measured $k$ values. Annual carbon outgassing for modelling with groundwater data yielded 34.9 kg C $m^{-2}$ $yr^{-1}$ when normalised to the stream surface area and was in the range between 32.7 and 42.9 kg C $m^{-2}$ $yr^{-1}$ of annual values obtained by equations for $k$ in Raymond et al. (2012).

Results indicate that fluxes according to Raymond et al. (2012) yielded reasonable estimates for annual fluxes, but showed deficiencies in reproducing temporal variability (Fig. 2). The discrepancies are likely due to uncertainties in the selection of an appropriate $k$ value in temporal highly variable headwater streams where variables such as slope, flow velocity, stream depth, and discharge are insufficient to predict the variability of $k$.

Moreover, for $CO_2$ loss ($[DIC]_{ex}$) in mg C $L^{-1}$ and daily average discharge in L $s^{-1}$, a negative concentration-discharge relationship was observed (Fig. 4). Higher $pCO_2$, modelled soil $pCO_2$ and modelled $CO_2$ loss occurred during low flow, when relative contributions of $CO_2$-enriched groundwaters to stream waters were high. A similar concentration-discharge

relationship was observed in a peatland stream, where deep soil and groundwater were considered as major $CO_2$ sources (Dinsmore and Billett, 2008).

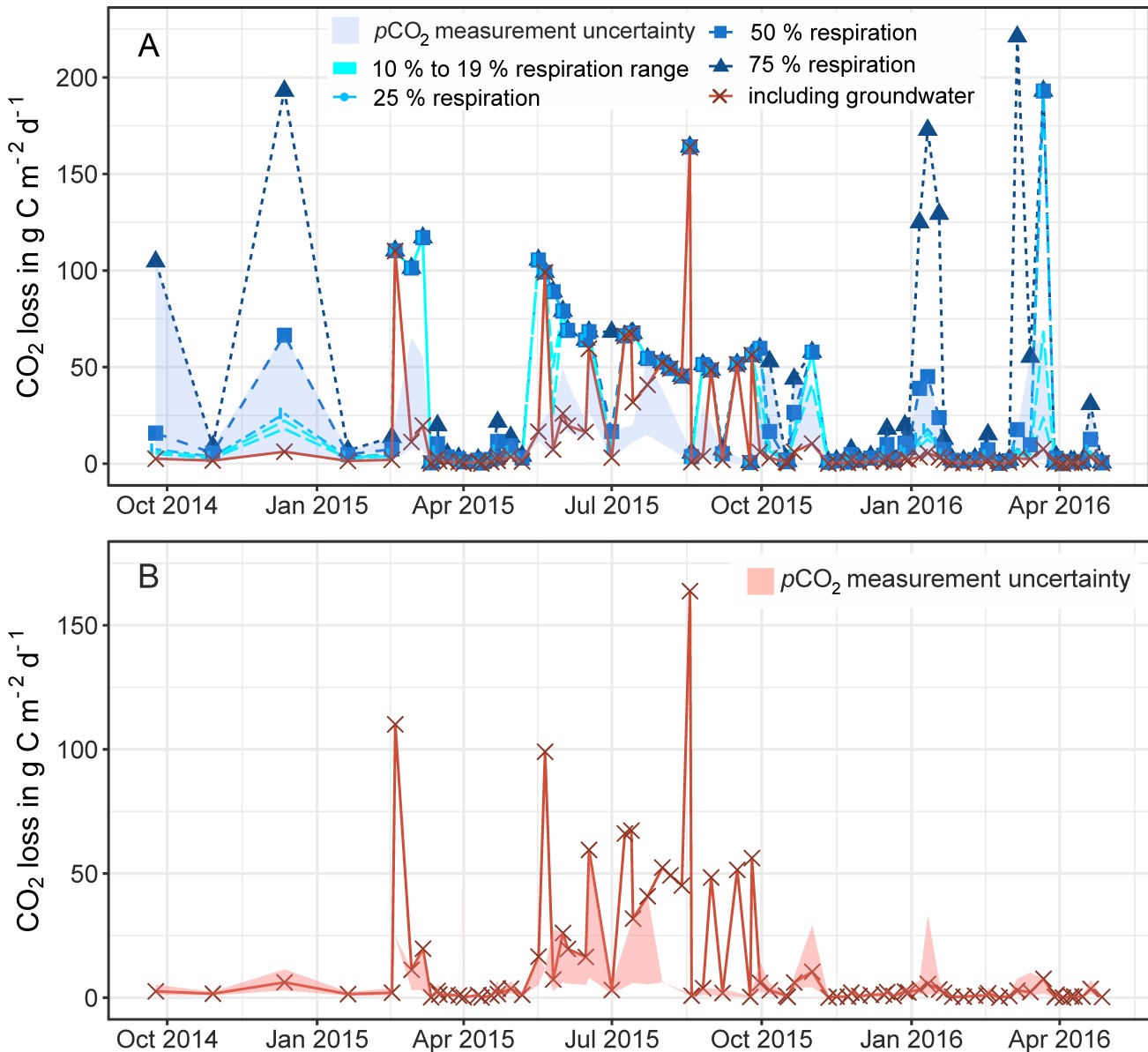

Figure 3: Modelled carbon dioxide loss via outgassing from the stream to the atmosphere upstream of the Uhlirska catchment outlet (UHL) based on the model by Polsenaere and Abril (2012) for proportion river respiration ($R$) between 10 and 75 % (A) and modified with measured groundwater $\delta^{13}C_{DIC}$ (A and B). The areas shaded in blue (A) and red (B) indicate uncertainties in calculation of $p$CO$_2$ with ±21 %. Convergence criteria were not fulfilled for data points that lie outside the shaded area (see section 3.1).

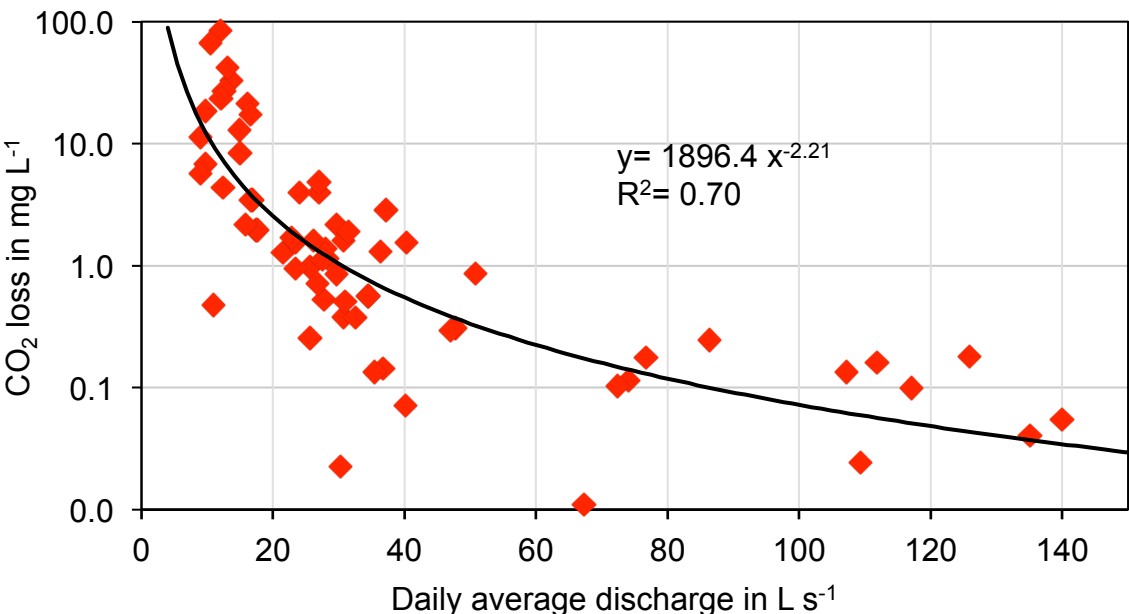

**Figure 4: Modelled CO$_2$ loss via outgassing upstream of the measurement point (UHL) versus daily average discharge. Modelling results correspond to modelling with groundwater input.**

### 3.5 Export of DIC proportions

Measured DIC together with modelled CO$_2$ outgassing from the stream surface to the atmosphere allowed the calculation of exported DIC species distributions. On an annual basis, the DIC proportions of HCO$_3^-$ export, CO$_2$* (i.e. the sum of CO$_{2(aq)}$ and H$_2$CO$_3$) export, and CO$_2$ outgassing had averages of 0.6, 1.2, and 7.1 to 10.3 mg C L$^{-1}$ for CO$_2$ outgassing with $R = 10$ and 19 % (Table 3). This corresponds to the relative amounts of approximately 6:12:82 % HCO$_3^-$ export, CO$_2$* export, and CO$_2$ outgassing with respect to total inorganic carbon loss from the Uhlirska catchment outlet.

For modelling with groundwater, the relative proportion of CO$_2$ outgassing to annual DIC export was even higher. With concentrations of 0.6, 1.2, and 10.3 mg C L$^{-1}$ of HCO$_3^-$ export, CO$_2$* export, and CO$_2$ outgassing they were approximately 5:10:85 %. These values are comparable to relations found by other studies (Billett et al., 2004;Johnson et al., 2008;Davidson et al., 2010;Polsenaere and Abril, 2012) and largely differ from findings from a headwater stream in karstic bedrock, where <30 % of DIC was outgassed as CO$_2$ (Lee et al., 2017). The determined DIC proportions suggest that a very large proportion (>80 %) of DIC entering the headwater stream in the silicate Uhlirska catchment rapidly outgasses as CO$_2$.

### 4 Conclusions

Numerous studies have demonstrated the importance of CO$_2$ outgassing from rivers and streams to the atmosphere on global carbon budgets and pointed out the restricted data on headwater catchments (Raymond et al., 2013;Lauerwald et al.,

2015;Aufdenkampe et al., 2011). Here we present a new study that successfully applied, validated, and modified the $CO_2$ degassing model by Polsenaere and Abril (2012) to a carbonate-free headwater catchment.

The modified isotope model was able to reproduce logical seasonal patterns of soil $pCO_2$ with a high variability of soil $pCO_2$ and $CO_2$ outgassing. It showed increased fluxes in summer and during snowmelt 2015. Modelled $CO_2$ losses also negatively
correlated with stream discharge. Results indicate maximum values of $CO_2$ outgassing from the stream to the atmosphere shortly before snowmelt. Modelled annual $CO_2$ outgassing was comparable to results obtained with model equations to calculate gas transfer velocities according to Raymond et al. (2012). However, results of the modified streamCO$_2$-DEGAS model showed larger variability, which indicates its potential to assess temporal $CO_2$ dynamics.

The model sensitivity to changing parameters of streamCO$_2$-DEGAS model, such as the proportion of river respiration ($R$),
in situ $pCO_2$ and $\delta^{13}C_{DIC}$, was high. This indicates that the potential to assess temporal variations becomes compromised by a larger potential for errors. This was particularly the case during summer.

Because of its decreased uncertainty, future $CO_2$ modelling would further benefit from direct $pCO_2$ measurement instead of its calculation. Such direct in situ methods include submerged infrared gas analysis (Johnson et al., 2010), equilibrator systems (Polsenaere et al., 2013), off-axis integrated cavity output spectrometer combined with a gas analyser (Gonzalez-
Valencia et al., 2014) and non-dispersive infrared sensors inside floating chambers (Bastviken et al., 2015;Lorke et al., 2015).

We circumvent the uncertainty of the respiration parameter ($R$) by incorporating wetland groundwater $\delta^{}C_{DIC}$ into the model. By using groundwater data, the modelling results have shown substantial improvements when compared to modelling without groundwater data. We therefore stress the importance of adding more accurate groundwater measurements to such
studies. This implies installation of more test wells in headwater catchments or the alternative use of local spring water as a proxy for groundwater. In addition, modelling on higher temporal resolution, particularly at the beginning of snowmelt, is needed to better reproduce dynamics and quantities of $CO_2$ outgassing.

**Acknowledgements**

This work was financially supported by the German Research Foundation (DFG, project 718 BA 2207/10-1). The authors
want to thank Pierre Polsenaere and Gwenael Abril for advice and sharing their Matlab code.
The authors declare no conflict of interest.

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
