# Peer review of "Groundwater data improve modelling of headwater stream CO2 outgassing with a stable DIC isotope approach"

_Biogeosciences, 2017_

## Referee Comment (RC1) · Anonymous Referee #1 · 27 Oct 2017

General comments: The authors of this manuscript apply an already existing model to assess the CO2 loss from a small silicate based watershed in Czech Republic. The novelty resides in the fact that their study is simplified by the fact that the watershed drains silicate rocks only and they use real data for the groundwater end-member instead of using literature values. The results from this manuscript are important since there are very few reliable data of CO2 emissions from first order streams to estimate global CO2 emissions from aquatic systems. I thus recommend the publication of this manuscript provided that the authors address the following specific comments.

Specific comments: P.6 L. 3-4. The authors mention that the d13C-DIC data have

been normalized to the VPDB scale by assigning +1.95 to NBS19 and -46.6 to LSVEC. These are solid carbonate materials which are very difficult to dissolve in water. It is thus unlikely that the authors have used these materials to normalize the raw data and if they did, they have likely broken the "identical treatment" principle. Here, the internal reference materials with their values and uncertainties need to be stated. P.6 L.6. The same applies for the d13C of POC. Also, I am sceptical that the Aurora-IRMS system was used to measure d13C of POC. Unless the authors mean TOC, the usual method is to use an EA online with an IRMS. The authors should clarify. P.6 L.15. In Equation 2, HCO3 was not measured. The authors should state how they have calculated HCO3 from the measured data and propagate their uncertainties. P.6 L.21. Even though the authors explain in depth their choice of an R value of 14% later in the discussion, they should explain here their choice of this value since there is no uncertainty associated to this value in the original cited paper. P.6 L.24. The authors state that they use d13C of POC instead of the d13 of SOM. In soils, DOC is often more important than POC and CO2 is more often linked to DOC than POC. Please explain why you use d13C of POC and not DOC or TOC. If d13C-POC = d13C-SOM, then explain. P.6 L.28. Why is the term R not part of the modelling with groundwater?

---

## Referee Comment (RC2) · Anonymous Referee #2 · 24 Nov 2017

Review of 'Groundwater data improve modelling of headwater stream CO2 outgassing with stable DIC isotope approach'

Summary: The manuscript by Dr. Marx and co-authors describes how the inclusion of GW d13C DIC data can improve the modelling of stream CO2 evasion estimates from small headwater streams, that contribute substantially to global freshwater evasion fluxes (Aufdenkampe et al., 2011, Raymond et al., 2013). For this, GW isotope data is incorporated into a stream degassing model that considers isotope fractionation (d13C) to estimate degassing. This approach is different than the most commonly applied method for estimating CO2 evasion. Commonly this flux is estimated

by combining measurements of pCO2 in streams with modelled and/or measured gas-transfer coefficients (k) (Dinsmore et al., 2013, Raymond et al., 2013, Wallin et al., 2013, Schelker et al., 2016). Thus the study aims to provide an alternative way to validate previous methods and to give a methodologically independent estimate of CO2 evasion form a small, acidic headwater stream.

Overall I find the work to be reasonably well performed and as such a possible basis for a publication. However, at present there are several points that hamper the story to me as a reader. Therefore I would ask the authors to rethink and revise the manuscript following the comments provided below.

Main comments:

1) There appears to be little data. The stream sampling covers only one single stream at very coarse temporal resolution (4-weeks interval for two years). Similarly, GW data is only from one single GW well (1) at three different depth. As a comparison, a similar study covering also POC and DIC stable isotope data (Polsenaere et al., 2013) measured 9 streams for one year at 2-weeks interval. That is ∼9 times more data than presented in this study. There is not much the authors can do about this now, but in case any other relevant datasets are available I would strongly encourage the authors to include any additional relevant material in the analysis.

2) The analysis clearly demonstrates the strong dependence of stream CO2 evasion estimates to the respiration of the stream ecosystem. Within the analysis a wide range of R-values is used; scenarios cover a very large range from 14 to 75%. As such, the methods for estimating stream metabolism from dissolved oxygen dynamics are well established and developed (Odum, 1956, Fisher & Likens, 1973, Demars et al., 2015) and some formulations have even explicitly considered the potential for GW inflows (Hall & Tank, 2005). The relevant measurement can for example be done by a dissolved oxygen logger for ∼1000 USD (for example from ONSET HOBO) that would have logged dissolved O2 dynamics continuously. Unfortunately, it appears no oxygen

measurements were applied in parallel to the C-isotope sampling that would allow an estimation of ecosystem respiration for this specific stream. Here I would at least expect that a literature analysis on streams with similar characteristics is performed to narrow down the possible range of respiration. However, in the paper only 'scenarios' for different contributions of ER to CO2 evasions are used and little is provided on this matter.

3) What is the exact topic of the paper? As such the material could provide a number of different angels. However, it appears as if the authors cannot really decide on which topic to choose. At present, the work is presented as a methodological advancement paper of the method of using stable C isotopes of GW to improve stream CO2 estimates. To really anchor the paper in the literature, an additional estimate of evasion fluxes by the more commonly used gas-transfer equation (Raymond et al., 2013) would have been required. Only then, one could conclude the true value of the new approach. Second, the paper is presently also placed as a contribution to the global literature on CO2 evasion estimates form headwater streams. For this, there is in fact very little data (only one stream) with a low temporal resolution (only one sample per month). Thus and despite the fact that this new approach is interesting and promising, I find only limited use of the manuscript in this context.

So in simple words: At present, I see the paper as 'neither apples nor oranges' and suggest the authors to revise the focus of the paper as good as it is possible.

4) The overall quality of the writing can be improved and is sometimes unprecise. There are many examples where statements are not clear, especially in the discussion. I have given some indications for these in my detailed comments below. Please rework the discussion. Also, make sure all the text is past-tense.

Minor comments:

Title: Good.

Abstract:

L14/15: appears contradicitive: they (small headwaters) contribute 36%... from all rivers and stream worldwide. How about changing 'all rivers and streams worldwide' with 'fluvial ecosystems on the globe' or something similar?

Introduction:

P1, L33: suggest to revise to "excluded streams of Strahler order below three" and leave the reference to Strahler out, as this is common knowledge, at least in hydrology. Also numbers below six should be spelled out (that's at least what I learned back in the days).

P2, L4: suggest to also reference (Schelker et al., 2016) for the statement on gas transfer velocities.

P2, L6: the Hotchkiss et al., reference does not appear to be a good reference here. It is a modeling paper that is based on a large number of streams.

P2, L11, Schelker et al., 2016 would again,fit well.

P2, L16: (Reichert et al., 2009) would be a good reference, as they provide estimates of the upstream length that significantly contributes to what is measured at the sampling location x donwstream.

P3, L1: "the aim was" - past tense.

P3, L2: Please add "and their stable isotope signature" to "measured groundwater contributions"

Methods:

L17: How is 'runoff intensity' defined and what unit would this variable have? This is not a standard term in hydrology! A sum of discharge? Or peak-discharge? Besides the need to clarify the term (or replacing it), I suggest to provide some numbers for this

in Table 1.

P5, L2: As pointed out in my main comment, it is unfortunate, that there is so little data. A monthly sampling for a stream with a nival flow regime, essentially means, that all relevant runoff episodes will, if at all, just be sampled by chance and then only one. If one is unlucky, there is not a single sample from the spring freshet. For me this is a major drawback of this dataset and of this study.

P5, L4: no need to cite anything here. Remove ref, or give the reader some information, why this ref is relevant here.

P5, L23: replace 'usually multiplied by 1000' with 'expressed as per mill'

P6, L7: The first sentence of this section is not well placed. This can go somewhere at the end of the section, as it is not very relevant. Instead a sentence or two that outline the model choices with a reasoning would be much more appropriate.

P6, L11: I am not convinced that the a-priori assumption that there is no relevant primary production is reasonably, as for example many northern streams have relevant PP, at least during summer lowflow, at the same time as the climate is similar to the stream of this study (Fisher & Likens, 1973). Anyhow, for the sake of the paper I suggest the authors to write something along the lines of 'For our study we assume that. . .' rather than claiming that there is no PP without providing explicit evidence from this specific stream.

P6, L22: The precise statement in the reference by Hotchkiss et al is:" Median internal $CO_2$ production increases from 14% (credible range = 10–19%) [. . .]". So as a matter of fact, this study does not give a single value, but rather a possible range of R. I suggest to consider this range, rather than a single value here and in other instances of the manuscript where R is discussed.

P7, L5 methods well described. Also great to see that one can extract a k-value that is then comparable to other studies.

P7, L12: Be sure to know the difference between a model parameter and an input variable and use the terms accordingly. At least in hydrological modelling, a measured value is not a parameter.

Results:

P8, L5: Please improve the writing here. If you begin a new section, add a topic sentence so that the reader knows what is described in the following paragraph. At present, this is just a horrible start of a results section. Similarly, please try to describe the results as such, and not just in which table/figure these are presented.

P8, L14: This sentence belongs to the methods, but not the results. Also, even if data normalized to the catchment area is interesting (and maybe better), most other studies on CO2 evasion have used the normalization by stream surface area. Thus I suggest to provide both these numbers (catchment and stream surface area normalized), so that the reader can compare with past results. Finally, I may add, that the argument for normalization by stream surface area has been, that remote sensing techniques can provide stream surface areas for large and remote areas and thus allow upscaling.

L20, averaged

P9, L1: interesting numbers!

P9, L4 please do not use any references in the results. Here only the data from this study is discussed, whereas any similarities and dissimilarities with other studies should be discussed din the discussion section.

L4 and 5: Is there any relationship of k600 and Q? Most studies assume this (Raymond et al., 2012), as higher Q means higher flow velocity and therefore turbulence and gas-exchange.

L6: Unfortunatly it is only here, that the reader understands that now some different models (or scenarios) are compared. Please add the purpose of these different scenarios/models to the methods section (see earlier comment).

L13, please be precise. Concentration of what? As a reader I only know what relationship this is, after looking at the figure!

L15: revise to 'does not follow this proposed relationship'. The observation as such is very interesting.

L16: again, no references nor comparisons within the results!

Discussion:

Please begin the discussion with describing the key result in a larger context... "This study shows/demonstrates...!" to create a red line for the forthcoming discussion. At present there is no red line here.

P12, L2: This first sentence is a prime example for my criticisms of unprecise writing. Have Polsanere and Abril really conclusively shown for this(!) catchment that initial pCO2 represents soil pCO2? This is how I read this sentence. Instead of making such bold statements, the authors should discuss, why they believe this is the case...

L: 8 and following: I agree with the enhanced soil respiration, which is a function of temperature and humidity. However, the sentence: "The main reason for higher soil pCO2 are larger contributions of CO2-enriched GW to stream water..." does not make sense! How would GW flowing into the stream affect soil pCO2 in a positive way, meaning increasing it?

L20-24: There is in fact some literature that has raised this point: (Pacific et al., 2008, Boodoo et al., 2017)

P13, L3: Good english writing means that place and time are placed at the end of the sentence, and in the order given before. Here (and in some other instances) this is not the case.

L8-10: This is the core results of the study! Please present this somewhere more prominent, rather than here, in the middle of the discussion!

[Figure]

L17 and following: These comparisons are good and interesting. How about a tbale that compares you findings on K600 with other studies. The advantage would be that the reader would actually get to see the other values and not just a 'in the range' statement. Obviously such a literature overview should only on stream of the same stream order, and somewhat similar characteristics.

On a similar matter and also concerning Figure 4: Here a comparison with other studies on a per-area (stream surface) would be great, as stream surface area and water volume increase with increasing discharge, so that the observed pattern of decreasing $CO_2$ loss per volume may be countered on a per area.

Conclusions:

P13, L25

L27: 'the Uhlriska' is redundant here and can be removed.

L30: The point that snowmelt puts out more $CO_2$ would be much clearer, if the per stream area values would be compared (see earlier comment)

P14, L7 and following: I don't really understand this... If the isotopes give such wonderful new possibilities, why are the authors then arguing that more chamber measurements should be done? Besides the fact that these have their own problems, the best would probably be chambers with C-isotopes, as it is already used in some terrestrial systems using a laser ring down spectroscopy analyzer. Please revise or remove. The results are not suggesting to do more bastviken-style chamber work!

Figures:

Figure 1: I can't really see the large map of Germany, Poland etc. lines too thin.

Figure 2 and 3: Essentially these two show the same thing: A timeseries of the different model scenarios (different Rs and the model with GW) as well as the respective uncertainties. Thus I suggest to merge the figures into one. Uncertainty ranges can

then be shown by background shading of different pattern and/or color.

Other comments: F2 caption: 'do not apply to'. Also, remove last sentence, as redundant, especially if figures are merged.

Fig4: A nice figure. Maybe add a little information, on the 'one outlier', that is the event with the highest flow either in the caption or directly in the figure (using an arrow). Also, how about plotting an exponential function to the data (maybe with the outlier removed), as this relationship seems pretty strong. Next, is CO2 loss defined as a term? I may have overlooked it, but its important that the reader knows if this is equal to CO2 evasion. Finally, please see my earlier comment, on the unit of 'loss' and how this pattern may change, if one takes a look at the entire stream reach.

Tables:

T1: Add runoff seasonality during the study years (see earlier comment). Also, stream length is 2.0km here, whereas it 2100m in the text.

T3: caption: these are not parameter, but results! Please revise. One option would eb to write 'DIC partitioning according to the . . . model'

Supplementary:

Fig.S1 and S2. These show some of the nice data of this work! Please compress these (for example discharge can be shown in all graphs in grey in the background) and present them as full figures in the manuscript. No reason to hide them.

TS1: please add a note on missing values

References:

Aufdenkampe AK, Mayorga E, Raymond PA et al. (2011) Riverine coupling of biogeochemical cycles between land, oceans, and atmosphere. Frontiers in Ecology and the Environment, 9, 53-60.
Boodoo KS, Trauth N, Schmidt C, Schelker J, Battin TJ (2017) Gravel bars are sites of increased CO2 outgassing in stream corridors. Scientific Reports, 7, 14401.

Demars BOL, Thompson J, Manson JR (2015) Stream metabolism and the open diel oxygen method: Principles, practice, and perspectives. Limnology and Oceanography: Methods, 13, 356-374.

Dinsmore KJ, Billett MF, Dyson KE (2013) Temperature and precipitation drive temporal variability in aquatic carbon and GHG concentrations and fluxes in a peatland catchment. Global Change Biology, 19, 2133-2148.

Fisher SG, Likens GE (1973) Energy Flow in Bear Brook, New Hampshire: An Integrative Approach to Stream Ecosystem Metabolism. Ecological Monographs, 43, 421-439.

Hall RO, Tank JL (2005) Correcting whole-stream estimates of metabolism for groundwater input. Limnology and Oceanography: Methods, 3, 222-229.

Odum HT (1956) Primary production in flowing waters. Limnol. Oceanogr, 1, 102–117.

Pacific VJ, Mcglynn BL, Riveros-Iregui DA, Welsch DL, Epstein HE (2008) Variability in soil respiration across riparian-hillslope transitions. Biogeochemistry, 91, 51-70.

Polsenaere P, Savoye N, Etcheber H, Canton M, Poirier D, Bouillon S, Abril G (2013) Export and degassing of terrestrial carbon through watercourses draining a temperate podzolized catchment. Aquatic Sciences, 75, 299-319.

Raymond PA, Hartmann J, Lauerwald R et al. (2013) Global carbon dioxide emissions from inland waters. Nature, 503, 355-359.

Raymond PA, Zappa CJ, Butman D et al. (2012) Scaling the gas transfer velocity and hydraulic geometry in streams and small rivers. Limnology & Oceanography: Fluids & Environments, 2, 41-53.

Reichert P, Uehlinger U, Acuña V (2009) Estimating stream metabolism from oxygen

concentrations: Effect of spatial heterogeneity. Journal of Geophysical Research: Bio-geosciences, 114, G03016.

Schelker J, Singer GA, Ulseth AJ, Hengsberger S, Battin TJ (2016) CO2evasion from a steep, high gradient stream network: importance of seasonal and diurnal variation in aquatic pCO2and gas transfer. Limnology and Oceanography, 61, 1826-1838.

Wallin MB, Grabs T, Buffam I, Laudon H, Ågren A, Öquist MG, Bishop K (2013) Eva-sion of CO2from streams - The dominant component of the carbon export through the aquatic conduit in a boreal landscape. Global Change Biology, 19, 785-797.

---

## Author Comment (AC1) · 21 Dec 2017

*On behalf of all authors, I thank the referee for the evaluation and constructive comments. In the text and tables below, responses to the points raised by the referee are answered in green and italic.*

**Response to Anonymous Referee #1:**

General comments: The authors of this manuscript apply an already existing model to assess the CO2 loss from a small silicate based watershed in Czech Republic. The novelty resides in the fact that their study is simplified by the fact that the watershed drains silicate rocks only and they use real data for the groundwater end-member instead of using literature values. The results from this manuscript are important since there are very few reliable data of CO2 emissions from first order streams to estimate global CO2 emissions from aquatic systems. I thus recommend the publication of this manuscript provided that the authors address the following specific comments.

*We thank the reviewer for the positive and constructive evaluation of our manuscript.*

| Comment by reviewer | Answer/ action by authors |
|---|---|
| P.6 L. 3-4. The authors mention that the d13C-DIC data have been normalized to the VPDB scale by assigning +1.95 to NBS19 and -46.6 to LSVEC. These are solid carbonate materials which are very difficult to dissolve in water. It is thus unlikely that the authors have used these materials to normalize the raw data and if they did, they have likely broken the "identical treatment" principle. Here, the internal reference materials with their values and uncertainties need to be stated. | *The reviewer is correct. The description of the normalization procedure for the Aurora 1030W analyzer was not correct. We did not use NBS19 and LSVEC here, as NBS19 would not dissolve in water (LSVEC would). We deleted that sentence and apologize for the confusion.* *In contrast, the normalization procedure of the Aurora TIC-TOC analyzer relies on $CO_2$ liberated from organic substances that readily dissolve in water, such as sugar, which in turn were normalized to VPDB by EA-IRMS via international reference materials USGS 40, USGS41 and IAEA CH-6. The precision of the internal control standard (C3-sugar) is stated in the text. For details on this instrumental setup we refer to St Jean (2003) (for the reference see the manuscript) that decribes the system in detail.* *Further, the referee is correct that we did not apply the "identical treatment" principle in a strict sense here. This is simply because of the facts that no international DIC reference material exist and that for this peripheral the normalization procedure for $\delta^{13}C$-DIC and $\delta^{13}C$-DOC typically relies on $CO_2$ that is completely liberated from organic reference materials.* *We revised the method description and provide more details on the analytical procedure.* |
| P.6 L.6. The same applies for the d13C of POC. Also, I am sceptical that the Aurora-IRMS | *The reviewer is correct and we apologize for the confusion. The methods section is about DIC and only the last sentence refers* |

| | |
|---|---|
| system was used to measure d13C of POC. Unless the authors mean TOC, the usual method is to use an EA online with an IRMS. The authors should clarify. | *to POC. For the POC method we refered to Barth et al. (2017).*
 *The POC was indeed measured on an EA-IRMS system and we now provide the full details on this analytical procedure in the methods section:*

 *"POC samples were analyzed for $\delta^{13}C_{POC}$ using a Costech Elemental Analyzer (ECS 4010; Costech International, Pioltello, Italy; now NC Technologies, Bussero, Italy) coupled in continuous flow mode to a Thermo Scientific Delta V plus IRMS. The data sets were corrected for linearity and instrumental drift during the run. Values were normalized for carbon to VPDB by analyses of internal reference materials ($C_4$ sugar and KHP) that were calibrated directly versus USGS-40 and USGS-41 (L-glutamic acid). Assigned values to USGS-40 and USGS-41 were −26.39‰ and +37.63‰ for $\delta^{13}C$, respectively. For precision and accuracy two laboratory standards (acetanilide and tartaric acid) were measured in each run. Precision, defined as the standard deviation of the control standard was better than 0.1‰ (1s) for $\delta^{13}C_{POC}$."* |
| P.6 L.15. In Equation 2, HCO3 was not measured. The authors should state how they have calculated HCO3 from the measured data and propagate their uncertainties. | *$HCO_3^-$ was calculated with measured DIC, pH and dissociation constants with equations 1 and 2 being inserted in equation 3 and solved for $HCO_3^-$ (Dickson et al. 2007).*

 $K1 = \frac{[H^+][HCO_3^-]}{[H_2CO_3]} \rightarrow [H_2CO_3] = \frac{[H^+][HCO_3^-]}{[K1]}$ *(1)*

 $K2 = \frac{[H^+][CO_3^{2-}]}{[HCO_3^-]} \rightarrow [CO_3^{2-}] = \frac{[K2][HCO_3^-]}{[H^+]}$ *(2)*

 $DIC = [H_2CO_3] + [HCO_3^-] + [CO_3^{2-}]$ *(3)*

 *Dissociation constants are temperature-dependent and were calculated according to Plummer & Busenberg (1982). The uncertainty of the calculation depends on the measurement uncertainties of DIC, pH and T.*
 *The largest uncertainty is usually caused by pH measurements as they are on a logarithmic scale. For instance, the uncertainty for pH is typically smaller than ±0.1 causes a maximum uncertainty of ±21 % for $pCO_2$. We consider this as a worst-case scenario that is also indicated in Figure 2 and 3.*
 *We added these aspects and the relevant* |

| | |
|---|---|
| | *literature to the manuscript.* |
| P.6 L.21. Even though the authors explain in depth their choice of an R value of 14% later in the discussion, they should explain here their choice of this value since there is no uncertainty associated to this value in the original cited paper. | *In a study about 187 streams and rivers the original paper (Hotchkiss et al. 2015) indicated a range between 10 and 19 % with 14 % being a median value for headwater streams. In our calculations we worked with this median value, but the revised version of the manuscript we now work with the range between 10 and 19 % to better reflect possible respiration.* |
| P.6 L.24. The authors state that they use $\delta^{13}C$ of POC instead of the d13 of SOM. In soils, DOC is often more important than POC and $CO_2$ is more often linked to DOC than POC. Please explain why you use d13C of POC and not DOC or TOC. If d13C-POC = d13C-SOM, then explain. | *In our opinion, POC is the best representative of soil organic matter in the catchment. It is the material that is closest to the original plant and soil material.*
 *We did not choose DOC, because it is known to have recalcitrant phases that are poorly decomposed (Cauwet and Sidorov 1996; Laudon et al. 2011). This would introduce a non-representative value for organic matter input.* |
| P.6 L.28. Why is the term R not part of the modelling with groundwater? | *The original model computes the isotopic composition of soil $CO_2$ from the isotopic composition of soil organic matter ($\delta^{13}C$-SOM) (see eq. (1) in Polsenaere & Abril, 2012). In unsaturated soils the respiration related fractionation is large and variable (up to 4-5 ‰) and thus Polsenaere & Abril included R in eq. (1). We skipped this step with its high uncertainty by using groundwater data instead.*
 *During respiration in waterlogged soils and groundwaters the isotopic fractionation of $CO_2$ is much smaller (near 1 ‰, O'Leary, 1984). Thus we included 1 ‰ fractionation to the groundwater input.*
 *We changed the text passage in the manuscript.* |

*References*

*Cauwet, G., and I. Sidorov (1996), The biogeochemistry of Lena River: Organic carbon and nutrients distribution, Mar. Chem., 53(3-4), 211-227, doi: 10.1016/0304-4203(95)00090-9.*

*Dickson, A. G. S., C.L.; Christian, J.R. (Ed.) (2007), Guide to best practices for ocean $CO_2$ measurements. PICES Special Publication 3, 191 pp.*

*Laudon, H., M. Berggren, A. Agren, I. Buffam, K. Bishop, T. Grabs, M. Jansson, and S. Kohler (2011), Patterns and Dynamics of Dissolved Organic Carbon (DOC) in Boreal Streams: The Role of Processes, Connectivity, and Scaling, Ecosystems, 14(6), 880-893, doi: 10.1007/S10021-011-9452-8.*

---

## Author Comment (AC2) · 22 Dec 2017

*On behalf of all authors, I thank the referee for the evaluation and constructive comments. In the text and tables below, responses to the points raised by the referes are answered in green and italic.*

**Response to Anonymous Referee #2:**

Summary: The manuscript by Dr. Marx and co-authors describes how the inclusion of GW d13C DIC data can improve the modelling of stream CO2 evasion estimates from small headwater streams, that contribute substantially to global freshwater evasion fluxes (Aufdenkampe et al., 2011, Raymond et al., 2013). For this, GW isotope data is incorporated into a stream degassing model that considers isotope fractionation (d13C) to estimate degassing. This approach is different than the most commonly applied method for estimating CO2 evasion. Commonly this flux is estimated by combining measurements of pCO2 in streams with modelled and/or measured gastransfer coefficients (k) (Dinsmore et al., 2013, Raymond et al., 2013, Wallin et al., 2013, Schelker et al., 2016). Thus the study aims to provide an alternative way to validate previous methods and to give a methodologically independent estimate of $CO_2$ evasion form a small, acidic headwater stream.

Overall I find the work to be reasonably well performed and as such a possible basis for a publication. However, at present there are several points that hamper the story to me as a reader. Therefore I would ask the authors to rethink and revise the manuscript following the comments provided below.

*The authors thank the reviewer for the positive and constructive evaluation of the manuscript.*

Main comments:

1) There appears to be little data. The stream sampling covers only one single stream at very coarse temporal resolution (4-weeks interval for two years). Similarly, GW data is only from one single GW well (1) at three different depth. As a comparison, a similar study covering also POC and DIC stable isotope data (Polsenaere et al., 2013) measured 9 streams for one year at 2-weeks interval. That is _9 times more data than presented in this study. There is not much the authors can do about this now, but in case any other relevant datasets are available I would strongly encourage the authors to include any additional relevant material in the analysis.

*The sampling for isotopes in the investigated catchment was performed over a period of 20 months. We noticed that the study by Polsenaere et al. (2012) used more data points (in terms of time and space), however, in our study of the Uhlirska headwater catchment only one representative stream exists in the catchment. For logistic and scientific reasons we decided to increase the sampling time at cost of a lower frequency (every month) to cover a longer time span (20 month; almost two years). Measuring longer time spans will allow for the consideration of seasonal changes in the model approach. We considered this as more important than a higher temporal resolution of the data.*
*Groundwater data is from three different wells in the wetland. The wells have different depths with 2.7, 3.7 and 5.2 m below ground level. Calculated averages for the three wells are used for modeling. We assume that these wells do best reflect carbon inputs to the stream. Groundwater wells in headwater catchments are largely uncommon and thus to provide original groundwater data that are often not available in other studies are the best we can offer. In the conclusion chapter, we argue that more representative groundwater data are certainly necessary to strengthen the model.*

2) The analysis clearly demonstrates the strong dependence of stream CO2 evasion estimates to the respiration of the stream ecosystem. Within the analysis a wide range of R-

values is used; scenarios cover a very large range from 14 to 75%. As such, the methods for estimating stream metabolism from dissolved oxygen dynamics are well established and developed (Odum, 1956, Fisher & Likens, 1973, Demars et al., 2015) and some formulations have even explicitly considered the potential for GW inflows (Hall & Tank, 2005). The relevant measurement can for example be done by a dissolved oxygen logger for _1000 USD (for example from ONSET HOBO) that would have logged dissolved O2 dynamics continuously. Unfortunately, it appears no oxygen measurements were applied in parallel to the C-isotope sampling that would allow an estimation of ecosystem respiration for this specific stream. Here I would at least expect that a literature analysis on streams with similar characteristics is performed to narrow down the possible range of respiration. However, in the paper only 'scenarios' for different contributions of ER to CO2 evasions are used and little is provided on this matter.

*In the streamCO$_2$-DEGAS model the term R stands for the proportion of CO$_2$ coming from respiration in water along the entire river course (Polsenaere & Abril, 2012, eq. (1)). This is different from ecosystem respiration and cannot be quantified by dissolved oxygen methods; R has to be considered together with isotope changes caused by diffusion. If the CO$_2$ stems from respiration in the unsaturated zone, its isotope composition changes by up to 4.4 ‰ by diffusion. If the CO$_2$ originates from respiration in water, its isotope composition is identical to soil organic matter and diffusion does not apply. Therefore the term R indicates the relative contributions of respiration from saturated and unsaturated zones. In our model, R was varied between 10 and 75 %.*

3) What is the exact topic of the paper? As such the material could provide a number of different angels. However, it appears as if the authors cannot really decide on which topic to choose. At present, the work is presented as a methodological advancement paper of the method of using stable C isotopes of GW to improve stream CO2 estimates. To really anchor the paper in the literature, an additional estimate of evasion fluxes by the more commonly used gas-transfer equation (Raymond et al., 2013) would have been required. Only then, one could conclude the true value of the new approach. Second, the paper is presently also placed as a contribution to the global literature on CO2 evasion estimates form headwater streams. For this, there is in fact very little data (only one stream) with a low temporal resolution (only one sample per month). Thus and despite the fact that this new approach is interesting and promising, I find only limited use of the manuscript in this context. So in simple words: At present, I see the paper as 'neither apples nor oranges' and suggest the authors to revise the focus of the paper as good as it is possible.

*We agree with the reviewer that the estimation of CO$_2$ fluxes by gas-transfer equations is a common approach. However, in our study we decided not to include CO$_2$ fluxes calculated by the gas transfer velocity k (Raymond et al., 2013) for comparison with our model results, because the use of k values is not recommended for headwater streams and exhibit very large uncertainties (Huotari et al. 2013, Wallin et al. 2011). For instance, comparing our results to those calculated according to Raymond et al. 2013 yielded large differences with an average of about 50 % with a range from 4 to 87 % for different dates. We think these discrepancies are due to uncertainties in the selection of an appropriate k value for heterogeneous headwater streams where tubulances, flow velocities and stream morphologies rapidly change. Thus, these estimates should not be the basis to evaluate the value of our approach. Nevertheless we included Fluxes determined with k values in the revised manuscript.*
*Overall, our study is a methological manuscript, in which we present, model and test actual data from a real-world headwater catchment. It contributes to the global CO$_2$ evasion discussion by providing an advanced method, however, we are not claiming to improve global datasets with this study alone.*

4) The overall quality of the writing can be improved and is sometimes unprecise. There are many examples where statements are not clear, especially in the discussion. I have given

some indications for these in my detailed comments below. Please rework the discussion. Also, make sure all the text is past-tense.

*The authors appreciate the reviewer's detailed and constructive comments that we will address in detail in the table below.*

Minor comments:
Title: Good.

*Below we reply on each specific comment.*

| Comment by reviewer | Answer/ action by authors |
|---|---|
| **Abstract:**
L14/15: appears contradicitive: they (small headwaters) contribute 36%... from all rivers and stream worldwide. How about changing 'all rivers and streams worldwide' with 'fluvial ecosystems on the globe' or something similar? | *We changed the text as recommended.* |
| **Introduction:**
P1, L33: suggest to revise to "excluded streams of Strahler order below three" and leave the reference to Strahler out, as this is common knowledge, at least in hydrology. Also numbers below six should be spelled out (that's at least what I learned back in the days). | *Done.* |
| P2, L4: suggest to also reference (Schelker et al., 2016) for the statement on gas transfer velocities. | *We added the reference.* |
| P2, L6: the Hotchkiss et al., reference does not appear to be a good reference here. It is a modeling paper that is based on a large number of streams. | *We replaced the reference by Stets et al. (2017).* |
| P2, L11, Schelker et al., 2016 would again,fit well. | *We added the suggested reference.* |
| P2, L16: (Reichert et al., 2009) would be a good reference, as they provide estimates of the upstream length that significantly contributes to what is measured at the sampling location x donwstream. | *We added the reference as recommended by the referee.* |
| P3, L1: "the aim was" - past tense. | *We changed the text as recommended.* |
| P3, L2: Please add "and their stable isotope signature" to "measured groundwater contributions" | *Done.* |
| **Methods:**
L17: How is 'runoff intensity' defined and what unit would this variable have? This is not a standard term in hydrology! A sum of discharge? Or peak-discharge? Besides the need to clarify the term (or replacing it), I suggest to provide some numbers for this in Table 1. | *Sentence is changed to: „However, during snowmelt the monthly average runoff doubles (>50 L s$^{-1}$) compared to the other months' runoff (Table 1)."* |
| P5, L2: As pointed out in my main comment, it is unfortunate, that there is so little data. A | *The referee is correct. The data is insufficient to cover the relevant runoff* |

| | |
|---|---|
| monthly sampling for a stream with a nival flow regime, essentially means, that all relevant runoff episodes will, if at all, just be sampled by chance and then only one. If one is unlucky, there is not a single sample from the spring freshet. For me this is a major drawback of this dataset and of this study. | *episodes. In the conclusions we now explicitly argue for higher resolution data in future studies.* |
| P5, L4: no need to cite anything here. Remove ref, or give the reader some information, why this ref is relevant here. | *We deleted the reference.* |
| P5, L23: replace 'usually multiplied by 1000' with 'expressed as per mill' | *Due to comments of referee #1 the methods section was thouroughly revised. We replaced the term as suggested.* |
| P6, L7: The first sentence of this section is not well placed. This can go somewhere at the end of the section, as it is not very relevant. Instead a sentence or two that outline the model choices with a reasoning would be much more appropriate. | *We shifted the first sentence to the end of the section and start section 2.3 with an explanation of model choices:*

*"At the Uhlirska headwater catchment the original streamCO$_2$-DEGAS model was applied to calculate stream CO$_2$ outgassing for different scenarios with varying values of river respiration (R) with 10, 19, 25, 50 and 75 % to test the model sensitivity to these values. In a second approach we modified the streamCO$_2$-DEGAS model as follows: instead of soil organic matter ($\delta^{13}C_{SOM}$) we used groundwater $\delta^{13}C_{DIC}$ data to better constrain initial CO$_2$ values and to reduce the model uncertainty."* |
| P6, L11: I am not convinced that the a-priori assumption that there is no relevant primary production is reasonably, as for example many northern streams have relevant PP, at least during summer lowflow, at the same time as the climate is similar to the stream of this study (Fisher & Likens, 1973). Anyhow, for the sake of the paper I suggest the authors to write something along the lines of 'For our study we assume that. . .' rather than claiming that there is no PP without providing explicit evidence from this specific stream. | *We changed the text accordingly: "The Uhlirska catchment meets the assumption of the streamCO$_2$-DEGAS model with (i) stream waters being acidic with pH values between 4.7 and 6.8 (Table 2), and for our study we assumed that (ii) waters in the stream are unproductive. This means that secondary processes such as photosynthesis by algae or biofilms and DOC degradation to CO$_2$ are neglected. This is a plausible assumption because high runoff and short residence times often leave insufficient time for substantial degradation of DOC (Raymond et al., 2016;Catalan et al., 2016). However, the potential of temperature-dependent aquatic bio- and photodegradation (Demars et al., 2011;Moran and Zepp, 1997) particularly during summer cannot be entirely excluded in the Uhlirska catchment." This was also considered in the discussion section.* |
| P6, L22: The precise statement in the reference by Hotchkiss et al is:" Median internal CO2 production increases from 14% (credible range | *In the revised manuscript we did the modeling with R= 10 % and R= 19% to reflect the range instead of the median* |

| | |
|---|---|
| = 10–19%) [. . .]”. So as a matter of fact, this study does not give a single value, but rather a possible range of R. I suggest to consider this range, rather than a single value here and in other instances of the manuscript where R is discussed. | *value suggested by Hotchkiss et al. 2015 (also see our comment above).* |
| P7, L5 methods well described. Also great to see that one can extract a k-value that is then comparable to other studies. | *We appreciate the given credit.* |
| P7, L12: Be sure to know the difference between a model parameter and an input variable and use the terms accordingly. At least in hydrological modelling, a measured value is not a parameter. | *The nomenclature is now consistent and changed throughout the text.* |
| **Results:**
P8, L5: Please improve the writing here. If you begin a new section, add a topic sentence so that the reader knows what is described in the following paragraph. At present, this is just a horrible start of a results section. Similarly, please try to describe the results as such, and not just in which table/figure these are presented. | *We thank the reviewer for pointing this out. We went through the entire manuscript and hopefully improved this aspect according to the reviewer's suggestion.*
*This section now starts with:*
*"We were able to estimate DIC export with modelled $CO_2$ outgassing and calculated lateral export of $HCO_3^-$, $CO_2$\* as well as of total DIC (Table 3). These data cover a period of 20 months and measurements took place at the catchment outlet, whereas modelling results relate to $CO_2$ outgassing between the stream source and the catchment outlet (UHL)."* |
| P8, L14: This sentence belongs to the methods, but not the results. Also, even if data normalized to the catchment area is interesting (and maybe better), most other studies on CO2 evasion have used the normalization by stream surface area. Thus I suggest to provide both these numbers (catchment and stream surface area normalized), so that the reader can compare with past results. Finally, I may add, that the argument for normalization by stream surface area has been, that remote sensing techniques can | *We moved the sentence to the methods and changed it according to:*
*"To allow for inter-catchment comparisons the carbon losses were normalised to the stream surface area. In addition, to avoid often imprecise stream lengths and surface areas, the carbon losses were also normalised to the catchment area."*
*The mean stream surface area of 1.49 $km^2$ (calculated from stream geometry) was added to Table 1. Fluxes normalized to stream surface area were added to the results section.* |
| L20, averaged | *We think „mean values" would be the more precise term here. We changes "averages" to "mean values".* |
| P9, L1: interesting numbers! | *We agree. This means that a large amount of inorganic carbon is outgassed in the form of $CO_2$.* |
| P9, L4 please do not use any references in the results. Here only the data from this study is discussed, whereas any similarities and dissimilarities with other studies should be discussed din the discussion section. | *The reviewer is correct and we apologize for this inconsistency of the structure. We moved all statements with references from the results to the discussion section.* |

| | |
|---|---|
| L4 and 5: Is there any relationship of k600 and Q? Most studies assume this (Raymond et al., 2012), as higher Q means higher flow velocity and therefore turbulence and gasexchange. | *The k600 values obtained with our model correlation versus Q has an $R^2$ of 0.6 and for modeling with groundwater data the $R^2$ is 0.4 (see Figures below). However, the k600 values from our model depend on discharge. The reason is that k600 is calculated from F which is the flux that is calculated by multiplying the modeled concentration by discharge.*

*For this reason we prefer Figure 4, which displays model results ($CO_2$ loss in mg/L) versus discharge rather then a plot of Q vs. k600.* |
| L6: Unfortunatly it is only here, that the reader understands that now some different models (or scenarios) are compared. Please add the purpose of these different scenarios/ models to the methods section (see earlier comment). | *The fact that two different approaches were used is now outlined at the beginning of section 2.3 in the revised manuscript (see comment above).* |
| L13, please be precise. Concentration of what? As a reader I only know what relationship this is, after looking at the figure! | *We changed the sentence to:*
*"Moreover, for $CO_2$ loss in mg $L^{-1}$ and daily average discharge in $L\ s^{-1}$ a negative concentration-discharge relationship was observed (Fig. 4)."* |
| L15: revise to 'does not follow this proposed relationship'. The observation as such is very interesting. | *We moved the sentence to the discussion section.* |
| L16: again, no references nor comparisons within the results! | *We deleted the sentence from the results section.* |
| **Discussion:**
Please begin the discussion with describing the key result in a larger context. . . "This study shows/demonstrates. . .!" to create a red line for the forthcoming discussion. At present there is no red line here. | *The revised the discussion. The section now starts with:*
*"Our study shows that the uncertainty of the respiration parameter (R) can be circumvented by incorporating wetland groundwater $\delta^{13}C_{DIC}$ into the streamCO2-DEGAS model. In a first step the original $CO_2$-DEGAS model was run with an R value of 25, 50 and 75 % (Fig. 2, Supplement Table S2). By doing so, the initial soil $pCO_2$ had to be extremely large (> 150000 ppmV) for selected months to reach the convergence of both $\delta^{13}C_{DIC}$ and $pCO_2$ at the same iteration (Polsenaere and Abril, 2012)…"* |
| P12, L2: This first sentence is a prime example for my criticisms of unprecise writing. Have Polsanere and Abril really conclusively shown for this(!) catchment that initial pCO2 represents soil pCO2? This is how I read this sentence. Instead of making such bold statements, the authors should discuss, why they believe this is the case. . . | *We rephrased our statement to:*
*"The streamCO2-DEGAS model assumes that the modelled initial $pCO_2$ represents soil $pCO_2$ (Polsenaere and Abril, 2012). For the Uhlirska catchment this would mean that soil $pCO_2$ values ranged between…"* |
| L: 8 and following: I agree with the enhanced soil respiration, which is a function of | *The referee correctly identified an inconsistent line of argument. We* |

| | |
|---|---|
| temperature and humidity. However, the sentence: "The main reason for higher soil pCO2 are larger contributions of CO2-enriched GW to stream water. . . " does not make sense! How would GW flowing into the stream affect soil pCO2 in a positive way, meaning increasing it? | *corrected the statement to the fact that higher respiration rates exist in summer due to warmer temperatures.* |
| L20-24: There is in fact some literature that has raised this point: (Pacific et al., 2008, Boodoo et al., 2017) | *These helpful references were incorporated into the revised version: "Although this type of respiration was not measured directly, we can assume a large potential of respiration in waters of the organic-rich wetland and of riparian soils with peak values during late summer and early fall (Pacific et al., 2008). In addition, respiration in gravel bar waters along the stream (Boodoo et al., 2017) can lead to an exceedance of 19 % for R along the Cerna Nisa stream."* |
| P13, L3: Good english writing means that place and time are placed at the end of the sentence, and in the order given before. Here (and in some other instances) this is not the case. | *We changed the sentences to: "Total alkalnity was low with 218 and 196 µmol L$^{-1}$ in the Uhlirska catchment during June and July 2015. In contrast, $\delta^{13}C_{DIC}$ values and in situ $pCO_2$ were increased with −15.0 and −15.2 ‰ as well as 2120 and 1910 ppmV, respectively."* |
| L8-10: This is the core results of the study! Please present this somewhere more prominent, rather than here, in the middle of the discussion! | *We agree. We included this point to the beginning of the section:* "Our study shows that the uncertainty of the respiration parameter (R) can be circumvented by incorporating wetland groundwater $\delta^{13}C_{DIC}$ into the streamCO$_2$-DEGAS model…" *(Also see comment above).* |
| L17 and following: These comparisons are good and interesting. How about a tbale that compares you findings on K600 with other studies. The advantage would be that the reader would actually get to see the other values and not just a 'in the range' statement. Obviously such a literature overview should only on stream of the same stream order, and somewhat similar characteristics.

On a similar matter and also concerning Figure 4: Here a comparison with other studies on a per-area (stream surface) would be great, as stream surface area and water volume increase with increasing discharge, so that the observed pattern of decreasing CO2 per volume may be countered on a per area. | *A table with corresponding values was added (see below).*

We agree that a comparison with other studies would be beneficial. However, the term of CO$_2$ loss is model-specific and the model would then first have to be applied to other catchments. By calculating fluxes normalized to stream surface areas (e.g. mg m$^{-2}$ month$^{-1}$) CO$_2$ loss would have to be multiplied by runoff. Thus this flux term would not be independent of runoff and a plot against runoff not advisable (see |

| | |
|---|---|
| | *answer above). No changes done.* |
| **Conclusions:**
P13, L25
L27: 'the Uhlriska' is redundant here and can be removed. | *We removed 'the Uhlriska' at the end of the sentence.* |
| L30: The point that snowmelt puts out more $CO_2$ would be much clearer, if the per stream area values would be compared (see earlier comment) | *We agree with the referee and will add a time series figure of $CO_2$ flux normalized to the area to the Figures we transferred from the supplement to the main text.* |
| P14, L7 and following: I don't really understand this... If the isotopes give such wonderful new possibilities, why are the authors then arguing that more chamber measurements should be done? Besides the fact that these have their own problems, the best would probably be chambers with C-isotopes, as it is already used in some terrestrial systems using a laser ring down spectroscopy analyzer. Please revise or remove. The results are not suggesting to do more bastviken-style chamber work! | *We advocated direct $pCO_2$ measurements here, because we think that they provide better inputs than its calculation via DIC, pH and T.*
*Chambers are mentioned here because they can be used to directly measure (time-weighted average) $pCO_2$ values. In our opinion this determination of stream $pCO_2$ has smaller uncertainties than its calculation via DIC, T and pH.* |
| **Figures:**
Figure 1: I can't really see the large map of Germany, Poland etc. lines too thin. | *We shaded the country areas to highlight the borders. Please see the revised version of Fig.1 below.* |
| Figure 2 and 3: Essentially these two show the same thing: A timeseries of the different model scenarios (different Rs and the model with GW) as well as the respective uncertainties. Thus I suggest to merge the figures into one. Uncertainty ranges can then be shown by background shading of different pattern and/or color. | *We updated Figure 2 with R = 10 – 19 %. Due to the complexity of the figure we suggest to keep Figure 2 and 3 separated and instead combine them into a single Figue 2 with the revised figures as Figures 2 A and 2 B (see below).* |
| Other comments: F2 caption: 'do not apply to'. Also, remove last sentence, as redundant, especially if figures are merged. | *Done.* |
| Fig4: A nice figure. Maybe add a little information, on the 'one outlier', that is the event with the highest flow either in the caption or directly in the figure (using an arrow). Also, how about plotting an exponential function to the data (maybe with the outlier removed), as this relationship seems pretty strong.

Next, is CO2 loss defined as a term? I may have overlooked it, but its important that the reader knows if this is equal to CO2 evasion.

Finally, please see my earlier comment, on the unit of 'loss' and how this pattern may change, if one takes a look at the entire stream reach. | *The outlier relates to a model result, where no convergence was reached and the model largely overestimates CO2 loss. Thus it does not show $CO_2$ release related to runoff generation.*
*We decided to plot the figure without the outlier and included a regression line. Please find the revised Figure 4 below.*

*The term of $CO_2$ loss basically is the model result and equals the loss of $CO_2$ via outgassing upstream of the measurement point.*
*We revised the figure caption.*

*For the answer regarding the unit of 'loss' see above.* |

| | |
|---|---|
| **Tables:**
T1: Add runoff seasonality during the study years (see earlier comment). Also, stream length is 2.0km here, whereas it 2100m in the text. | *The reviewer is right. We changed the stream length to 2.1 km as it is the correct value.*
*Regarding runoff seasonality we added: "Snowmelts in 2015 and 2016: Q> 50 L s⁻¹"* |
| T3: caption: these are not parameter, but results! Please revise. One option would be to write 'DIC partitioning according to the . . . model' | *We changed the caption as recommended by the reviewer.* |
| **Supplementary:**
Fig.S1 and S2. These show some of the nice data of this work! Please compress these (for example discharge can be shown in all graphs in grey in the background) and present them as full figures in the manuscript. No reason to hide them. | *They now appear in the main text as Figure 3.* |
| TS1: please add a note on missing values | *We marked the missing values by '*' and added a note beneath the table.* |

*References*

*Catalan, N., R. Marce, D. N. Kothawala, and L. J. Tranvik (2016), Organic carbon decomposition rates controlled by water retention time across inland waters, Nat. Geosci., 9(7), 501, doi: 10.1038/Ngeo2720.*

*De Fátima F. L. Rasera, M., M. Victoria R. Ballester, A. V. Krusche, C. Salimon, L. A. Montebelo, S. R. Alin, R. L. Victoria, and J. E. Richey (2008), Estimating the surface area of small rivers in the southwestern amazon and their role in CO2 outgassing, Earth Interactions, 12(6), doi: 10.1175/2008EI257.1.*

*Demars, B. O. L., J. R. Manson, J. S. Olafsson, G. M. Gislason, R. Gudmundsdottir, G. Woodward, J. Reiss, D. E. Pichler, J. J. Rasmussen, and N. Friberg (2011), Temperature and the metabolic balance of streams, Freshwater Biol., 56(6), 1106-1121, doi: 10.1111/j.1365-2427.2010.02554.x.*

*Harman, C., M. Stewardson, and R. DeRose (2008), Variability and uncertainty in reach bankfull hydraulic geometry, J. Hydrol., 351(1-2), 13-25, doi: 10.1016/j.jhydrol.2007.11.015.*

*Huotari, J., S. Haapanala, J. Pumpanen, T. Vesala, and A. Ojala (2013), Efficient gas exchange between a boreal river and the atmosphere, Geophys. Res. Lett., 40(21), 5683-5686, doi: 10.1002/2013GL057705.*

*Leopold, L. B., and T. J. Maddock (1953), Hydraulic geometry of stream channels and some physiographic implications. U. S. Geological Survey Professional Paper 252, 55 p.*

*Moran, M. A., and R. G. Zepp (1997), Role of photoreactions in the formation of biologically labile compounds from dissolved organic matter, Limnol. Oceanogr., 42(6), 1307-1316.*

*Raymond, P. A., J. E. Saiers, and W. V. Sobczak (2016), Hydrological and biogeochemical controls on watershed dissolved organic matter transport: pulse-shunt concept, Ecology, 97(1), 5-16, doi: 10.1890/14-1684.1.*

*Raymond, P. A., C. J. Zappa, D. Butman, T. L. Bott, J. Potter, P. Mulholland, A. E. Laursen, W. H. McDowell, and D. Newbold (2012), Scaling the gas transfer velocity and hydraulic geometry in streams and small rivers, Limnol. Oceanogr.-Fluids Environ., 41-53, doi: 10.1215/21573689-1597669.*

*Stets, E. G., D. Butman, C. P. McDonald, S. M. Stackpoole, M. D. DeGrandpre, and R. G. Striegl (2017), Carbonate buffering and metabolic controls on carbon dioxide in rivers, Global Biogeochem. Cy., 31(4), 663-677, doi: 10.1002/2016gb005578.*

*Wallin, M. B., M. G. Oquist, I. Buffam, M. F. Billett, J. Nisell, and K. H. Bishop (2011), Spatiotemporal variability of the gas transfer coefficient ($K_{CO2}$) in boreal streams: Implications for large scale estimates of $CO_2$ evasion, Global Biogeochem. Cy., 25, doi: 10.1029/2010gb003975.*

*Figures*

[Figure]

*Figures are provided to support the answer to reviewer comment 'P9 L4 and L5' about the relationship between k600 and Q.*

[Figure]

*Figure 1: Location of the Uhlirska catchment and sampling sites modified from Sanda et al. (2014).*

[Figure]

*Figure 2: Modelled monthly carbon dioxide loss via outgassing from the stream to the atmosphere upstream of the Uhlirska catchment outlet (UHL) based on the model by Polsenaere and Abril (2012) for proportion river respiration (R) between 10 and 75 % (A) and modified with measured groundwater $\delta^{13}C_{DIC}$ (A and B). The areas shaded in blue (A) and red (B) indicate uncertainties in calculation of pCO$_2$ with ±21 %. Convergence criteria were not fulfilled for data points that lie outside the shaded area (see Sect. 4).*

[Figure]

*Figure 4: Modelled $CO_2$ loss via outgassing upstream of the measurement point (UHL) versus daily average discharge. Modelling results correspond to modeling with groundwater input.*

*Table 4. Gas transfer velocities (k600) normalized to a stream temperature of 20°C of low order streams.*

| Region | Stream order | $k_{600}$ in m d$^{-1}$ | Reference |
|---|---|---|---|
| Temperate streams (25°-50°) | - | 4.8 [b, c] | Aufdenkampe et al. (2011) |
| Uhlirska, Czech Republic | 1 | 4.9 [a, b] | This study |
| Rene, France | - | 2.9 [a] | Polsenaere and Abril (2012) |
| United States | <4 | 4.5 [b] | Butman and Raymond (2011) |
| Wiesent, Germany | - | 6.3 [a] | van Geldern et al. (2015) |
| Rappbode, Germany | - | 2.9 [a] | Halbedel and Koschorrek (2013) |
| Hassel, Germany | - | 2.4 [a] | Halbedel and Koschorrek (2013) |
| Zillierbach, Germany | - | 2.2 [a] | Halbedel and Koschorrek (2013) |
| Ochsenbach, Germany | - | 2.5 [a] | Halbedel and Koschorrek (2013) |
| Boreal and arctic streams (50°-90°) | - | 3.1 [b, c] | Aufdenkampe et al. (2011) |
| Québec, Canada | 1 | 0.6 [a] | Campeau et al. (2014) |
| Québec, Canada | 2 | 0.6 [a] | Campeau et al. (2014) |
| Québec, Canada | 3 | 0.5 [a] | Campeau et al. (2014) |
| Québec, Canada | 4 | 1.4 [a] | Campeau et al. (2014) |
| Alaska, United States | $\leqq$4 | 6.5 [a] | Crawford et al. (2013) |
| Sweden | 1 | 15.5 [a] | Humborg et al. (2010) |
| Sweden | 2 | 12.4 [a] | Humborg et al. (2010) |

[a] Mean values.
[b] Median values.
[c] Running waters in Aufdenkampe et al. (2011) have < 60 – 100 m width.

---

## Author Response (AR2)

**Editor comment:**

Dear Anne,
Thank you for your detailed author replies and revised manuscript, which has now been re-evaluated by one of the original reviewers. I agree with their conclusion that you have adequately handled the suggestions and comments raised by both reviewers. I am therefore pleased to accept your revised version for publication in BG, pending a minor suggestion from my side (DOC & d13C-DOC data, see comment below), related to
one of the questions by Referee #1:
Ref#1: P.6 L.24. The authors state that they use δ13C of POC instead of the d13 of SOM. In soils,DOC is often more important than POC and CO2 is more often linked to DOC than POC.Please explain why you use d13C of POC and not DOC or TOC. If d13C-POC = d13C-SOM, then explain.
Author Reply: In our opinion, POC is the best representative of soil organic matter in the catchment. It is the material that is closest to the original plant and soil material. We did not choose DOC, because it is known to have recalcitrant phases that are poorly decomposed (Cauwet and Sidorov 1996; Laudon et al. 2011). This would introduce a non-representative value for organic matter input.

-->I would argue here that there are similarly recalcitrant pools in the soil POC, I do not find this line of reasoning supported by strong arguments or data. Currently, the manuscript mentions that both DOC concentrations and d13C-DOC data were measured (described in the Methods section), but neither the DOC data nor the d13C-DOC are presented or discussed. At the minimum, the d13C-DOC data should be presented, and you could briefly discuss to which extent using d13C-DOC rather then d13C-POC would change the output. Secondly, since DOC data were measured it would be a small extra effort to compare the DIC fluxes (incl CO2 outgassing) to export fluxes of DOC, then at least these data add something extra (in case you decide this is not wanted, then remove these analyses from the Materials & Methods section).

Best regards
Steven Bouillon

**Response to Editor comment:**

Dear Steven,

thank you again for handling our manuscript and for your constructive comment. We chose POC according to Polsenaere and Abril (2012), who developed the model and used POC as input parameter. To be consistent with their model we also used POC in our study, and not DOC.

With respect to POC versus DOC representing SOM, the choice of POC is surely discussable. However, in our manuscript we modified the original model with groundwater input. In this step, SOM (either represented by POC or DOC) is not needed as input parameter anymore. We thus decided to remove DOC and δ13CDOC from the methods section (2.2). In conjunction, we also removed the argument about recalcitrant phases of DOC (section 2.3).

Sincerely

Anne Marx

[revised manuscript text omitted]